# Neural Active Learning Meets the Partial Monitoring Framework

Maxime Heuillet[1]        Ola Ahmad[1,3]        Audrey Durand[1,2]

[1]Université Laval, Canada
[2]Canada-CIFAR AI Chair, Mila, Canada
[3]Thales Research and Technology (cortAIx), Canada,

## Abstract

We focus on the online-based active learning (OAL) setting where an agent operates over a stream of observations and trades-off between the costly acquisition of information (labelled observations) and the cost of prediction errors. We propose a novel foundation for OAL tasks based on partial monitoring, a theoretical framework specialized in online learning from partially informative actions. We show that previously studied binary and multi-class OAL tasks are instances of partial monitoring. We expand the real-world potential of OAL by introducing a new class of cost-sensitive OAL tasks. We propose `NeuralCBP`, the first PM strategy that accounts for predictive uncertainty with deep neural networks. Our extensive empirical evaluation on open source datasets shows that `Neural-CBP` has competitive performance against state-of-the-art baselines on multiple binary, multi-class and cost-sensitive OAL tasks.

## 1 INTRODUCTION

In active learning [Cohn et al., 1994], an agent decides to query an expert to obtain labels on selected observations. This active acquisition of labels efficiently reduces the number of labelled observations needed to learn a task. Active learning therefore appears as a cost-effective solution for modern machine learning, which often relies on large volumes of labelled observations [Kusne et al., 2020].

In this work, we focus on the *online-based active learning* (OAL) setting for binary and multi-class classification tasks Beygelzimer et al. [2009]. The agent operates over a (possibly infinite) stream of observations. For each observation, the agent predicts the class and either decides to reveal its prediction or to query an expert to obtain the label. The OAL setting we consider differs from the *batch setting* where the agent gathers fixed-size batches of observations to label Saran et al. [2023], Amin et al. [2020]. In both the OAL and the batch-based settings, all decisions are irrevocable and associated with costs. The goal is to minimize the cumulative cost over the stream of decisions, by trading-off between the cost of obtaining new labels (*labeling complexity*) and the cost of prediction errors (*generalization performance*).

In the context of OAL for binary classification, the `Margin` strategy [Sculley, 2007] queries the expert when the prediction uncertainty is greater than a user-specified threshold. In contrast, with `Cesa` [Cesa-Bianchi et al., 2006], labelled observations are acquired proportionally to the global prediction error rate of the strategy. Both `Margin` and `Cesa` are specifically analyzed for the class of linear separators and are designed for binary tasks. More recent studies focused on multi-class OAL tasks. The `Gappletron` [van der Hoeven et al., 2021] leverages graph feedback, making it inherently multi-class. However, simialrly to `Cesa` and `Margin`, `Gappletron` is specifically analyzed for linear separators.

Modern applications of machine learning involve high-dimensional observations that require learning complex representations. As a result, `Neural` [Wang et al., 2021] and `ALPS` [DeSalvo et al., 2021] proposed multi-class OAL strategies based on deep neural networks. `Neural` and `ALPS` have been outperformed by `INeural` [Ban et al., 2022b], an improved and more practical version of the `Neural` strategy [Wang et al., 2021]. The current state-of-the-art, `Neuronal` [Ban et al., 2024], addresses scalability limitations of `INeural`, opening the door to using sophisticated neural architectures, such as convolutional neural networks.

In critical real-world applications, the costs of prediction errors from one class to another may vary significantly. Cost-sensitive OAL is studied for regression tasks [Cai et al., 2023], but classification tasks remain an open problem. Existing OAL strategies all assume *uniform costs*, i.e. prediction error and labeling costs are the same across all classes. This assumption is the core of the algorithmic design of ex-

isting approaches, making it challenging to extend them to cost-sensitive tasks. Motivated by the fact that cost-sensitive learning has fostered the adoption of supervised learning in real-world scenarios, such as learning from imbalanced data [Elkan, 2001], we address the following questions: *1) How to frame cost-sensitive OAL classification tasks? and, 2) Can we design a practical cost-sensitive OAL agent?*

**Contributions.** We tackle these questions from a novel perspective based on Partial Monitoring (PM) [Piccolboni et al., 2001, Bartók et al., 2014], a theoretical framework for online learning problems with partially informative actions. ① Connecting ideas in separate fields: We hypothesize and validate that we can establish a novel, non-trivial, connection between the field of active learning and the PM framework. ② Methodological: We show how partial monitoring reduces to existing binary and multi-class OAL tasks and enables the formulation of novel cost-sensitive OAL tasks. ③ Algorithmic: We propose NeuralCBP (Neural Confidence Bound Partial Monitoring), a partial monitoring (PM) strategy able to learn from neural networks. Existing PM strategies are limited to the linear [Heuillet et al., 2024] and logistic [Bartók et al., 2012a], which constitute a bottleneck towards the adoption of PM in practice. NeuralCBP presents algorithmic dynamics that differ from existing OAL strategies, which can be of independent interest to the OAL community. ④ Empirical: Our empirical evaluation shows that NeuralCBP competes with the current state-of-the-art in multiple binary and multi-class settings, and across various neural architectures. ⑤ Advocacy: Documented applied studies of PM are limited to synthetic experiments Singla et al. [2014], Kirschner et al. [2023], Heuillet et al. [2024]. Furthermore, PM is a field traditionally more supported by theoretical advances. Our work shows that PM is a promising framework that can be effectively applied in applied settings such as OAL. ⑥ Reproduciblity: To support adoption, our study is fully reproducible with open-source code and implementation details (see Appendix A).

# 2 BACKGROUND

A PM [Bartók et al., 2014] game is played between a learning agent and the environment over multiple rounds. The agent has a finite set of $N$ actions. The environment has a finite set of $M$ outcomes. The game is defined by a cost matrix $\mathbf{C} \in [0,1]^{N \times M}$ and a feedback matrix $\mathbf{H} \in \Sigma^{N \times M}$. The symbol space $\Sigma$ is arbitrary and is not necessarily numeric (i.e. could be symbols). Without loss of generality, we assume that feedback symbols associated with one action are distinct from those induced by the other actions. We note $\mathbf{c}_i$ the $i$-th row of the matrix $\mathbf{C}$. The same notation applies to matrix $\mathbf{H}$. A summary table of the important notations is reported in Table 1 in the Appendix.

## 2.1 DYNAMICS OF A GAME

Matrices $\mathbf{C}$ and $\mathbf{H}$ are revealed to the agent before the game begins. The horizon of rounds $T$ is unknown to the agent. At each round $t \in \{1, 2, \ldots, T\}$, the environment samples an observation $x_t \in \mathcal{X}$. We make no assumption regarding the sampling process of the observations. The environment then samples an outcome $y_t \in \{1, 2, \ldots, M\}$ from an *outcome distribution* that depends on $x_t$, and that is denoted $p(x_t) \in \Delta_M \subset \mathbb{R}^{M \times 1}$, where $\Delta_M$ is the $M$-dimensional probability simplex. We assume outcomes are sampled i.i.d with respect to the outcome distribution.

The agent observes $x_t$ and selects an action $i_t \in \{1, 2, \ldots, N\}$. Then, the agent then incurs a cost $\mathbf{C}[i_t, y_t]$ and receives a feedback symbol $h_t = \mathbf{H}[i_t, y_t]$, where $[i, y]$ denotes the element at row $i$ and column $y$. Therefore, costs and feedback symbols are deterministic elements of matrices $\mathbf{C}$ and $\mathbf{H}$ respectively. We emphasize that the agent only observes the feedback symbol $h_t$, with neither the outcome nor the cost being revealed.

The goal is to minimize the cost incurred in each round. This is achieved by selecting the action $i_t^\star$ that minimizes the expected cost for $x_t$, and is defined such that $i_t^\star = \text{argmin}_{1 \le i \le N} \mathbf{c}_i p(x_t)$. The performance of the agent is measured by the cumulative regret (to minimize) w.r.t. the optimal action strategy:

$$R(T) = \sum_{t=1}^{T} (\mathbf{c}_{i_t} - \mathbf{c}_{i_t^\star}) p(x_t), \qquad (1)$$

Eq. 1 scales sub-linearly with $T$ if the agent identifies the optimal action and commits to it over time. This requires to balance *exploration* (playing informative actions) and *exploitation* (minimizing per-round regret).

## 2.2 STRUCTURE OF A GAME

We now introduce two definitions to characterize the cost $\mathbf{C}$ and feedback $\mathbf{H}$ matrices of any PM game.

**Definition 2.1** (Cell decomposition, Bartók et al. [2012b])**.** The *cell* $\mathcal{O}_i$ is defined as the subspace in the probability simplex $\Delta_M$ such that action $i$ would be optimal. Formally, $\mathcal{O}_i = \{p \in \Delta_M, \forall j \in \{1, ..., N\}, (\mathbf{c}_i - \mathbf{c}_j) p \le 0\}$.

Based on the above definition, action $i$ is: (i) *dominated* if $\mathcal{O}_i = \emptyset$ (i.e. there is no outcome distribution s.t. the action is optimal); (ii) *degenerate* if it is not dominated and there exist action $k$ such that $\mathcal{O}_i \subsetneq \mathcal{O}_k$ (i.e. actions $i$ and $k$ are duplicates, both are jointly optimal under some outcome distribution); and (iii) *Pareto-optimal* otherwise. The set of Pareto-optimal actions is denoted $\mathcal{P}$.

For an action $i$, let $\sigma_i$ denote the number of unique feedback symbols on $\mathbf{h}_i$. Let $\Sigma_i = \{s_1, ..., s_{\sigma_i}\}$ denote the enumeration of symbols sorted by order of appearance in $\mathbf{h}_i$. Let $\pi_i(x_t) \in \Delta_{\sigma_i} \subset \mathbb{R}^{\sigma_i \times 1}$ denote the probability distribution of receiving each feedback symbol of action $i$ given $x_t$.

**Definition 2.2** (Signal matrix, Bartók et al. [2012b]). Given action $i$, the elements in the *signal matrix* $S_i \in \{0,1\}^{\sigma_i \times M}$ are defined as $S_i[u,v] = \mathbb{1}_{\{\mathbf{H}[i,v]=s_u\}}$.

**Property 2.3.** *The outcome distribution is connected to the feedback symbols distribution of action $i$ through the signal matrix $S_i$ with the relation $\pi_i(x_t) = S_i p(x_t)$.*

## 3 STREAM-BASED ACTIVE LEARNING AS A PARTIAL MONITORING GAME

OAL problems have been studied under varied feedback models, such as bandit Erez et al. [2024], Daniely and Helbertal [2013] and full information feedback Sakaue et al. [2024]. In this work, we propose a connection between OAL problems and the PM feedback model Bartók et al. [2014]. In particular, we leverage specific PM instances, known as *label-efficient* games, to capture OAL problems.

The original label-efficient game [Helmbold et al., 1997] is characterized by $N = 3$ actions (predict class 1, predict class 2, and query the expert), $M = 2$ outcomes (the ground-truth classes), and the following cost and feedback matrices:

$$\mathbf{C} = \begin{array}{c} \text{pred. class 1} \\ \text{pred. class 2} \\ \text{expert} \end{array} \begin{array}{cc} \text{class 1} & \text{class 2} \\ \left[ \begin{array}{cc} 0 & 1 \\ 1 & 0 \\ 1 & 1 \end{array} \right. \end{array} \Big], \mathbf{H} = \begin{array}{cc} \text{class 1} & \text{class 2} \\ \left[ \begin{array}{cc} \diamond & \diamond \\ \wedge & \wedge \\ \bot & \odot \end{array} \right]. \end{array}$$

For reproducibility, in Appendix A, we instantiate all the definitions presented above using the label-efficient game as an example. Several OAL studies on binary classification correspond to instances of the original label-efficient game [Cohn et al., 1994, Balcan et al., 2007, Beygelzimer et al., 2009].

Using the game theoretical definitions presented above and developed in Bartók et al. [2012b], we now introduce a generalization of this game to multi-class classification with possibly non-uniform costs and multiple experts.

**Generalized label-efficient game** The OAL classification task with $M$ classes and $E$ experts can be cast as a PM game with $N = M + E$ actions and $M$ outcomes. Without loss of generality, we assume that the actions $\{M+1, \ldots, M+E\}$ correspond to requesting a label from the $E \geq 1$ experts. All actions $i > M$ (i.e. actions associated with an expert) are *dominated* (see Def. 2.1) and admit $\sigma_i = M$ distinct symbols. The other actions $\{1, \ldots, M\}$ lead to a single feedback symbol, i.e. $\sigma_i = 1$.

The original label-efficient game corresponds to the single-expert ($E = 1$) binary task ($M = 2$) with a uniform cost matrix. In this work, we focus on single-expert multi-class ($M \geq 2$) games with a potentially non-uniform cost matrix. The multi-expert setting ($E > 1$) suggests that experts reveal the outcome (ground-truth label) with different stochasticity levels. Capturing this would require a different PM

setting where feedback is subject to noise, as studied by Kirschner et al. [2020, 2023].

**Connecting regret minimization and OAL** The cost matrix captures both the cost of querying an expert and the cost of prediction errors. The goal of PM agents is to minimize the regret (see Eq. 1), which corresponds to simultaneously minimizing the cost associated with label queries (label complexity) and the cost of prediction errors (generalization performance). This goal aligns with the objectives of established OAL methodologies [DeSalvo et al., 2021, Wang et al., 2021, Ban et al., 2022b, 2024].

## 4 THE NEURALCBP APPROACH

We now introduce NeuralCBP, a partial monitoring strategy able to learn from neural networks. While the emphasis of this study is on OAL classification tasks, NeuralCBP is a general PM strategy that can be applied to the broader diversity of PM games. Algorithm 1 displays the pseudo-code of NeuralCBP.

The proposed NeuralCBP builds upon CBP (Confidence Bound Partial Monitoring) methods Bartók et al. [2012b], which currently have limited practical potential due to linear [Heuillet et al., 2024] and logistic [Bartók et al., 2012a] model assumptions. For an observation $x_t$, the expected cost difference between two actions $i$ and $j$ is

$$\delta_{i,j}(x_t) = (\mathbf{c}_i - \mathbf{c}_j)p(x_t), \tag{2}$$

where $p(x_t)$ is unknown by definition of the PM game. Action $j$ is better than action $i$ when $\delta_{i,j}(x_t) > 0$.

**Definition 4.1** (Neighbors, Bartók et al. [2012b]). Two Pareto-optimal actions $i$ and $j$ are *neighbors* if $\mathcal{O}_i \cap \mathcal{O}_j$ is a $(M-2)$-dimensional polytope. The set of all neighbor pairs is denoted $\mathcal{N}$.

Two actions are neighbors when these actions can not be jointly optimal for a given outcome distribution. Therefore, given observation $x_t$, one only needs to compute $\delta_{i,j}(x_t)$ for neighbor pairs in $\mathcal{N}$ at round $t$, rather than for all the action pairs $\{i,j\}$ in the game [Bartók et al., 2012b].

### 4.1 OUTCOME AND FEEDBACK DISTRIBUTIONS

Recall that the agent does not observe the outcomes. Consequently, the agent cannot directly estimate the outcome distribution $p(x_t)$. As a result, estimating the expected cost difference $\delta_{i,j}(x_t)$ using Eq. 2 is not feasible in practice. This motivates additional definitions to estimate the expected loss difference in practice.

**Definition 4.2** (Observer set, Bartók et al. [2012b]). The set $V_{i,j}$ includes all actions that verify the relation $(\mathbf{c}_i - \mathbf{c}_j)^\top \in \oplus_{a \in V_{i,j}} \text{Im}(S_a^\top)$, where $\oplus$ corresponds to the direct sum.

**Algorithm 1:** NeuralCPB

---

**input :** $\mathcal{P}, \mathcal{N}$

Initialize $\theta_1, \theta_2$

$G_{a,t} = \lambda \mathbf{1}_{m+\sigma m}, \forall a \in \{1, \ldots, N\}$

**for** $t > N$ **do**

    Initialize $\mathcal{U}(t) \leftarrow \{\}$

    Receive observation $x_t$

    Get $\hat{\pi}(x_t)$ based on $f_1(x_t, \theta_1)$

    Get $w(x_t)$ based on $f_2(x_t, \theta_2)$

    **for** *each action-pair* $\{i, j\} \in \mathcal{N}$ **do**

        $\hat{\delta}_{i,j}(t) = \sum_{a \in V_{i,j}} v_{ija} \hat{\pi}_a(x_t)$

        $z_{i,j}(t) \leftarrow \sum_{a \in V_{i,j}} \|v_{ija}\|_2 w_a(x_t)$

        **if** $|\hat{\delta}_{i,j}(t)| \geq z_{i,j}(t)$ **then**

            Add $\{i, j\}$ to $\mathcal{U}(t)$

    Compute $D(t)$ based on $\mathcal{U}(t)$

    Obtain $\mathcal{P}(t)$ and $\mathcal{N}(t)$ based on $D(t)$

    $\mathcal{N}^+(t) \leftarrow \bigcup_{i,j \in \mathcal{N}(t)} N_{i,j}^+$

    $\mathcal{V}(t) \leftarrow \bigcup_{i,j \in \mathcal{N}(t)} V_{i,j}$

    Compute $\mathcal{R}(x_t)$

    $\mathcal{S}(t) \leftarrow \mathcal{P}(t) \cup \mathcal{N}^+(t) \cup (\mathcal{V}(t) \cap \mathcal{R}(x_t))$

    Play $a_t = \arg\max_{a \in \mathcal{S}(t)} W_a w_a(x_t)$

    Observe feedback $h_t$

    Update $\theta_1, \theta_2$ with Algorithm 2 (Appendix B)

    Update $G_{a_t,t}^{-1}$ (see Sherman et al. [1950])

---

**Definition 4.3** (Observer vectors, Bartók et al. [2012b]). Given action $a \in V_{i,j}$, the observer vector $v_{ija} \in \mathbb{R}^{\sigma_a}$ is selected to satisfy the relation $(\mathbf{c}_i - \mathbf{c}_j)^\top = \sum_{a \in V_{i,j}} S_a^\top v_{ija}$.

The set $V_{i,j}$ contains actions that induce informative feedback symbols about $\mathbf{c}_i - \mathbf{c}_j$. It is defined such that $\mathbf{c}_i - \mathbf{c}_j$ can be expressed as a linear combination of the signal matrix images of actions in $V_{i,j}$, with the observer vectors being the coefficients of the combination.

Combining Definitions 4.2 and 4.3 with Property 2.3 allows to express $\delta_{i,j}(x_t)$ as a function of the feedback distributions $\pi_a(x_t)$ of all actions $a \in V_{i,j}$:

$$\delta_{i,j}(x_t) = \sum_{a \in V_{i,j}} v_{ija}^\top \pi_a(x_t). \tag{3}$$

Consequently, on can compute the estimate $\hat{\delta}_{i,j}(x_t)$ using the feedback distribution estimates $\hat{\pi}_a(x_t)$ associated with the actions in $V_{i,j}$. Similarly, the uncertainty in the loss difference estimate $\hat{\delta}_{i,j}(x_t)$ is:

$$z_{i,j}(x_t) = \sum_{a \in V_{i,j}} \|v_{ija}\|_\infty w_a(x_t), \tag{4}$$

where $w_a(x_t)$ is the uncertainty on $\hat{\pi}_a(x_t)$ [Lienert, 2013]. Methods to compute $\hat{\pi}_a(x_t)$ and $w_a(x_t)$ depend on the setting considered, e.g. without side-observation Bartók et al.

[2012b], or with linear Heuillet et al. [2024], or logistic Bartók et al. [2014] side-information. We now present a method for the neural setting.

## 4.2 INFERENCE WITH NEURAL NETWORKS

The strategy INeural [Ban et al., 2022b] frames the OAL classification task under bandit feedback, where all actions are self-informative. As a result, INeural leverages the Explore-Exploit Networks (referred to as EENets) initially introduced for bandit feedback [Ban et al., 2022a]. The current state-of-the art in OAL (Neuronal [Ban et al., 2024]), is a follow-up strategy based on EENets that showcases the limitations of a bandit feedback structure in practice and highlights the importance of finding an adequate and general feedback structure for the diversity of OAL classification tasks. As a response, we extend EENets to address the exploration-exploitation trade-off in the general PM setting. This extension is non-trivial because the PM feedback/cost structure requires exploration techniques that go beyond techniques used in bandit feedback. Further technical differences between NeuralCBP and other EENets strategies are discussed in Section 6.

EENets comprise an *exploitation network* (denoted $f_1$) to estimate action values and an *exploration network* (denoted $f_2$) to quantify the uncertainty on the predictions of $f_1$.

**Exploitation network**    In the PM setting, the exploitation network $f_1$ predicts the feedback distributions required in Eq. 3. To instantiate efficiently the exploitation network $f_1$ in PM, we need to distinguish informative from non-informative actions.

**Definition 4.4** (Set of informative actions). The set of informative actions, $\mathcal{I} = \{i : i \in \{1, \ldots, N\}, \text{ and } \sigma_i \geq 2\}$, comprises all actions that induce at least two distinct feedback symbols.

**Definition 4.5** (Valid feedback symbols). The set of valid feedback symbols is noted $\Sigma_\mathcal{I} = \bigcup_{i \in \mathcal{I}} \Sigma_i$. The dimension of $\Sigma_\mathcal{I}$ is $\sigma = \sum_{i \in \mathcal{I}} \sigma_i$, which represents the total number of unique symbols induced by the actions in $\mathcal{I}$.

Current CBP strategies Bartók et al. [2012a], Lienert [2013], Heuillet et al. [2024] estimate the feedback distribution for all the actions of a game. However, Property 2.3 shows that for any uninformative action $i \notin \mathcal{I}$, the learned feedback distribution is always $\pi_i(x_t) = 1$. In some PM games, most of the actions are uninformative, as it is the case for *generalized label-efficient* games (presented in Section 3) where only expert actions are informative. Therefore, attributing learnable parameters to uninformative actions, as is done in current CBP strategies [Heuillet et al., 2024, Bartók et al., 2012a], turns out to be inefficient. In contrast, NeuralCBP

attributes learnable parameters only to the informative actions in the game (see Definition 4.4). Restricting learnable parameters to the subset of informative actions $\mathcal{I}$ is essential because `NeuralCBP` relies on the Explore-Exploit networks (`EENets`) that require a shared representation for the actions. Including non-informative actions would cause overfitting, unstable learning, and increased complexity.

**Definition 4.6** (Set of informative feedback distributions). The set of informative feedback distributions, denoted $\Pi(x_t) = \{\pi_i(x_t), i \in \mathcal{I}\}$, contains the (unknown) feedback distribution vectors of each informative action.

*Remark* 4.7. For practical purposes, remark that the set $\Pi(x_t)$ can be converted into a (flattened) $\sigma$-dimensional row vector. The conversion from a set to a flattened vector, and conversely from a flattened vector to a set, is possible because the cardinality $\sigma_i$ of each feedback distribution is known by definition of the PM game.

The network $f_1$, therefore, learns the flattened vector associated with $\Pi(x_t)$ and predicts the desired estimates $\hat{\pi}_i(x_t)$ for actions $i \in \mathcal{I}$. Network $f_1$ can be instantiated as a fully connected multi-layer perceptron of depth $L$ and width $m$:

$$f_1(x_t, \theta_1) = W_1^L \Psi(W_1^{L-1} \Psi(W_1^{L-2} \ldots \Psi(W_1^1 x_t))),$$

where $W_1^1 \in \mathbb{R}^{m \times d}$, $W_1^\ell \in \mathbb{R}^{m \times m}, 1 < \ell < L$, and $W_1^L \in \mathbb{R}^{\sigma \times m}$. The notation $\Psi(x_t) = \max(0, x_t)$ refers to the ReLU activation function. We use a multi-layer perceptron as an example, but we will see in the experiments that $f_1$ can instantiate other neural architectures.

The network $f_1$ is trained by performing stochastic gradient descent with the mean squared error $\mathcal{L}_1$ between the predictions of $f_1$ and the observed feedback symbols, defined as

$$\mathcal{L}_1(\theta_1) = \sum_{\tau=1, h_\tau \in \Sigma_\mathcal{I}}^{t-1} \frac{(f_1(x_\tau, \theta_1) - e(h_\tau))^2}{2},$$

where $e(\cdot)$ refers to a $\sigma$-dimensional one-hot encoding. Note that $\theta_1$ is updated based on the history of valid feedback symbols ($h_t \in \Sigma_\mathcal{I}$) and their associated observations ($x_t$).

Based on remark 4.7, the set of feedback distribution estimates over all the $N$ actions in the game is defined as

$$\hat{\pi}(x_t) = \{\hat{\pi}_i(x) \text{ if } i \in \mathcal{I}, [1] \text{ otherwise}, i \in \{1, \ldots, N\}\},$$

where $[1]$ is the feedback distribution vector of uninformative actions and $|\hat{\pi}(x)| = N$. The $i$-th element of $\hat{\pi}(x)$ is the feedback distribution estimate of action $i$. In Sec. 4.3, we will describe how $\hat{\pi}(x)$ is used to trade-off between exploration and exploitation.

**Exploration network**   The exploration network $f_2$ estimates the prediction error of network $f_1$. These estimates are used to quantify the uncertainty on the predictions of $f_1$, and they are used to compute the confidence formula defined in Eq. 4.

**Definition 4.8** (End-to-end embedding, Ban et al. [2024]). Given the exploitation network $f_1$ and an observation $x_t$, the end-to-end embedding is defined as

$$\phi(x_t) = \left[ \Psi(W_1^1 x_t)^\top, \text{vec}(\nabla_{W_1^L} f_1(x_t, \theta_1)^\top) \right] \in \mathbb{R}^{m+\sigma m},$$

where the first element is the output vector of the first layer of $f_1$ and the second element is the flattened (represented by operator `vec`) partial derivative of $f_1$ with respect to the parameters of the last layer. In practice, $\phi(x_t)$ is normalized by dividing all the elements by the $l_2$-norm of the vector.

To produce uncertainty estimates, the network $f_2$ learns the function $\Pi(x_t) - f_1(x_t, \theta_1)$. The network $f_2$ is instantiated as a multi-layer perceptron of depth $L$ and width $m$, which receives the end-to-end embedding $\phi(x_t)$:

$$f_2(x_t, \theta_2) = W_2^L \Psi(W_2^{L-1} \Psi(W_2^{L-2} \ldots \Psi(W_2^1 \phi(x_t)))),$$

where $W_2^1 \in \mathbb{R}^{m \times (m+\sigma m)}$, $W_1^\ell \in \mathbb{R}^{m \times m}, 1 < \ell < L$, and $W_1^L \in \mathbb{R}^{\sigma \times m}$. The weights $\theta_2$ of network $f_2$ are updated with stochastic gradient descent using the loss

$$\mathcal{L}_2(\theta_2) = \sum_{\tau=1, h_\tau \in \Sigma_\mathcal{I}}^{t-1} \frac{(f_2(x_\tau, \theta_2) - (e(h_\tau) - f_1(x_\tau, \theta_1)))^2}{2}.$$

*Remark* 4.9. The network $f_2$ is also based on the flattened vector representation of $\Pi(x_t)$. Therefore, the predictions of $f_2$ are $\sigma$-dimensional vectors. Since the number of symbols $\sigma_i$ is known for any action by definition of the PM game, we can convert the flattened prediction vector of $f_2$ into a set of vectors.

Given remark 4.9, let $w(x_t) = \{\max(\hat{w}_i(x_t)) \text{ if } i \in \mathcal{I}, 0 \text{ otherwise}, i \in \{1, \ldots, N\}\}$ denote the set of uncertainty estimates over all actions, where $|w(x_t)| = N$ and the notation $w_i(x_t)$ refers to the $i$-th element of $w(x_t)$. In other words, the uncertainty of an informative action corresponds to the maximum uncertainty value predicted by $f_2$ over the $\sigma_i$ symbols induced by action $i$, denoted $\max(\hat{w}_i(x_t))$. This can be thought of as the worst-case uncertainty for the informative action $i$. For uninformative actions, the uncertainty is 0 following the heuristic that $\pi_i(x_t) = [1]$ for $i \notin \mathcal{I}$.

### 4.3   EXPLORATION AND EXPLOITATION

By leveraging the PM-extended `EENets` mechanism, `NeuralCBP` can compute $\hat{\delta}_{i,j}(x_t)$ for all neighbor action pairs $\{i, j\} \in \mathcal{N}$ using the feedback distributions predicted by network $f_1$ (see Eq. 3). It can also compute uncertainty estimates $z_{i,j}(x_t)$ on $\hat{\delta}_{i,j}(x_t)$ using the uncertainties predicted by network $f_2$ (see Eq. 4).

Following the CBP methodology, `NeuralCBP` then separates low uncertainty from high uncertainty estimates of $\hat{\delta}_{i,j}(x_t)$ by using a *successive elimination* [Even-Dar et al.,

2002] criteria $|\hat{\delta}_{i,j}(x_t)| > z_{i,j}(x_t)$ for each action pair $\{i,j\} \in \mathcal{N}$. At round $t$, the pairs that verify the criteria are gathered in the set of confident pairs, denoted $\mathcal{U}(t)$. NeuralCBP leverages $\mathcal{U}(t)$ to compute a sub-space of the probability simplex $\Delta_M$, defined as $D(t) = \{p \in \Delta_M, \{i,j\} \in \mathcal{U}(t), \text{sign}(\hat{\delta}_{i,j}(x_t))(\mathbf{c}_i - \mathbf{c}_j)p > 0\}$. The set $D(t)$ thus contains all likely outcome distributions given the confident estimates of loss differences. The true (unknown) outcome distribution $p(x_t)$ is included with high confidence in the sub-space $D(t)$.

NeuralCBP then considers the set of *likely Pareto-optimal actions* $\mathcal{P}(t) \subseteq \mathcal{P}$ containing all Pareto-optimal actions $i \in \mathcal{P}$ such that their cell $\mathcal{O}_i$ intersects with the sub-space $D(t)$. Similarly, it considers the set of *likely neighbors* $\mathcal{N}(t) \subseteq \mathcal{N}$ containing all neighbor action pairs $\{i,j\} \in \mathcal{N}$ such that their common cell $\mathcal{O}_i \cap \mathcal{O}_j$ intersects with $D(t)$. When $\mathcal{P}(t)$ contains only one action, $\mathcal{N}(t)$ is automatically empty, and therefore NeuralCBP exploits. When $\mathcal{P}(t)$ contains more than one action, it explores.

**The selected action at round** $t$. Let $X_{i,t} = \{\phi(x_\tau)\}_{\tau=1, i_\tau=i}^{t-1}$ denote the history up to time $t$ (exclusively) of the observations embeddings under which action $i$ was selected, and let $G_{i,t} = \lambda I_d + X_{i,t} X_{i,t}^\top$ denote the associated Gram matrix. Let the notation $\|x\|_S^2 = x^\top S x$ denote the norm of vector $x$ weighted by some matrix $S$.

**Definition 4.10** (Underplayed actions, Heuillet et al. [2024]). The set of underplayed actions, $\mathcal{R}(x_t) = \{i \in \{1, \ldots, N\}$ s.t. $1/\|x_t\|_{G_{i,t}^{-1}}^2 < \eta_i f(t)\}$, contains actions that have been played less than some play rate function $f(t)$ weighted by a scalar $\eta_i > 0$. The quantity $1/\|x_t\|_{G_{i,t}^{-1}}^2$ is a pseudo-count of the number of times action $i$ was selected, weighted by the similarity between the current observation $x_t$ and the observations at previous selections of action $i$.

**Definition 4.11** (Neighborhood action set Bartók et al. [2012b]). The neighborhood action set of a neighbor pair $\{i,j\}$ is defined as $N_{i,j}^+ = \{k \in \{1, \ldots N\}, \mathcal{O}_i \cap \mathcal{O}_j \subseteq \mathcal{O}_k\}$. Note that $N_{i,j}^+$ naturally contains $i$ and $j$. If $N_{i,j}^+$ contains another action $k$, then $\mathcal{O}_k = \mathcal{O}_i$ or $\mathcal{O}_k = \mathcal{O}_j$ or $\mathcal{O}_k = \mathcal{O}_i \cap \mathcal{O}_j$.

NeuralCBP computes the *likely neighbor* action set $\mathcal{N}^+(t) = \bigcup_{\{i,j\} \in \mathcal{N}(t)} N_{i,j}^+$ based on the remaining action pairs (likely neighbors) in $\mathcal{N}(t)$. Similarly, the set of *likely observer actions* is defined as $\mathcal{V}(t) = \bigcup_{\{i,j\} \in \mathcal{N}(t)} V_{i,j}$.

The final set of actions considered by NeuralCBP at round $t$, denoted $\mathcal{S}(t)$, contains all potentially optimal actions (i.e. $\mathcal{P}(t) \cup \mathcal{N}^+(t)$) and all informative actions (i.e. $\mathcal{V}(t) \cup \mathcal{R}(x_t)$). From $\mathcal{S}(t)$, NeuralCBP selects the action with the greatest uncertainty weighted by $W_a = \max_{\{i,j\} \in \mathcal{N}} \|v_{ija}\|_\infty$, i.e. $a_t = \text{argmax}_{i \in \mathcal{S}(t)} W_a w_a(x_t)$.

# 5 EXPERIMENTS

We compare the empirical performance of NeuralCBP to state-of-the-art baselines on a set of binary, multi-class, and cost-sensitive OAL classification tasks. To evaluate the robustness across neural architectures, we conduct experiments with a multi-layer perceptron (MLP) and the convolutional architecture LeNet LeCun et al. [1998]. To our knowledge, our experiments are the first to evaluate OAL with a convolutional architecture. For reproducibility, we open-source the code base of NeuralCBP and the baselines. We also open-source the code base of PM-based OAL game environments. The code base is available on Github: https://github.com/MaxHeuillet/neuralCBPside.

**Datasets** For binary OAL tasks, we evaluate on **Adult** [Asuncion et al., 2007], **MagicTelescope** [Asuncion et al., 2007], and the **modified MNIST** [LeCun et al., 2010] (odds vs. even numbers) datasets. For multiclass OAL tasks, we consider **covertype** and **shuttle** from the UCI repository [Asuncion et al., 2007], **MNIST** [LeCun et al., 2010], **Fashion** [Xiao et al., 2017], and **CIFAR10** [Krizhevsky et al., 2009]. For each dataset, we put aside $15\%$ of the observations to create a separate fixed size *test set*, intended exclusively for evaluation of the generalization performance. We sample from the remaining observations a *deployment stream* that has a finite horizon of $T = 10k$ rounds. The OAL strategies acquire labelled data from the deployment stream. We run each experiment 25 times with different dataset splits for each run.

**Baselines** We compare NeuralCBP to six baselines. In the binary setting, we adapt the strategies Cesa and Margin, originally proposed for linear classifiers, to function with MLPs. We evaluate the multi-class state-of-the-art strategies INeural and Neuronal. Both INeural and Neuronal rely on a hyper-parameter $\gamma$ that influences the amount of exploration. We consider respectively two instances of each strategy: one with the hyper-parameter configuration specified in their official publications ($\gamma = 6$ for Neuronal and INeural) and one that we chose to induce less exploration ($\gamma = 3$ for Neuronal and INeural). We further discuss implementation details and hyper-parameters in Appendix C.

**Performance metrics** To characterize the performance on the deployment stream, we measure the *final cumulative regret* achieved at the end of the stream (Eq. 1), which has to be minimized. We also report the *win count*, i.e. the number of times a strategy achieves the lowest final regret at the end of the horizon across the 25 runs of each experiment. Lastly, we perform *one-sided Welch's t-tests* to asses if NeuralCBP's final regret distribution is significantly lower than the baselines.

To evaluate the generalization performance and account for possible data imbalance in the considered datasets, we mea-

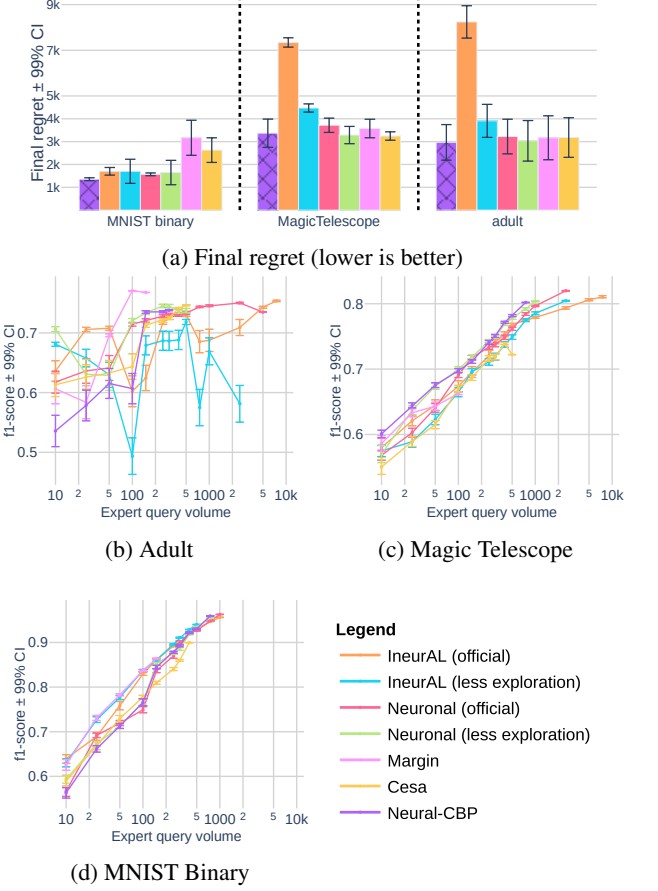

(a) Final regret (lower is better)

(b) Adult

(c) Magic Telescope

(d) MNIST Binary

**Legend**
- IneurAL (official)
- IneurAL (less exploration)
- Neuronal (official)
- Neuronal (less exploration)
- Margin
- Cesa
- Neural-CBP

Figure 1: Performance on binary OAL with MLP.

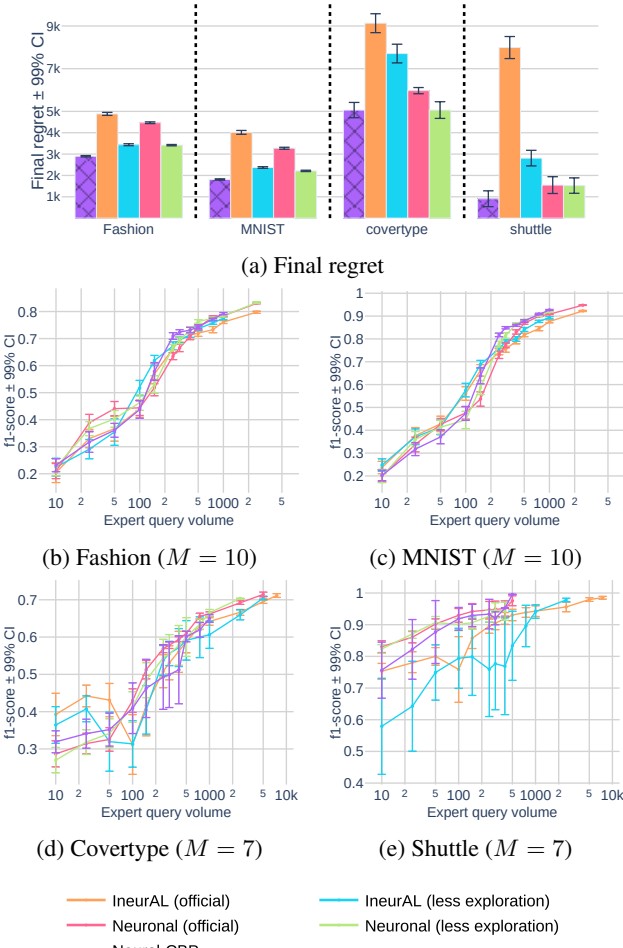

(a) Final regret

(b) Fashion ($M = 10$)

(c) MNIST ($M = 10$)

(d) Covertype ($M = 7$)

(e) Shuttle ($M = 7$)

- IneurAL (official)
- Neuronal (official)
- Neural-CBP
- IneurAL (less exploration)
- Neuronal (less exploration)

Figure 2: Performance on multi-class OAL with MLP.

sure the *weighted f1-score* on the test sets. We measure the weighted f1-score while considering different volumes of expert-labelled observations acquired on the deployment stream (10, 25, 50, 100, 150, 250, 300, 400, 500, 750, 1000, 2500, 5000, 7500, 9000). Fixing the number of labelled observations allows for a fair comparison between the strategies.

Lastly, we report the *average number of expert verifications* consumed by each strategy. It is important to note that, unlike previous experimental protocols [Ban et al., 2024, 2022b, DeSalvo et al., 2021], we do not set a maximum expert query budget. This choice aims to illustrate how each strategy effectively adapts its label complexity to the learning task. This choice reflects the realistic scenario where the optimal expert budget is unknown prior to deployment (as it largely depends on the dataset and type of architecture).

### 5.1 BINARY CASE

Figure 1a reports the final regrets achieved on the deployment stream. Numerical details are reported in Table 2 of Appendix C.1. `NeuralCBP` achieves the best final regret on the MNISTbinary and Adult datasets. On the MagicTelescope dataset, `NeuralCBP` achieves a final regret comparable `Neuronal`, `Cesa` and `Margin`, suggesting all these

strategies are close to the optimal solution on this dataset. We observe that the final regret of all the strategies is subject to high variance (see Fig. 1a) caused by variations in the task difficulty over the 25 dataset splits. As a result, it is insightful to assess the performance solely based on the average final regret metric. The win count reveals that over 25 trials, `NeuralCBP` outperforms the baselines with 10 wins on MNISTbinary, 14 wins on MagicTelescope, and 10 wins on Adult.

Figures 1b, 1c, and 1d display the weighted f1-score performance on the test sets for different volumes of expert queries. For each strategy, the f1-score curve stops at different expert-query volumes, which illustrates the *label complexity* of each approach. `NeuralCBP` exhibits a lower label query complexity (see Figures 1b and 1c) and achieves a f1-score performance that is comparable to the one achieved by the other baselines.

### 5.2 MULTI-CLASS CASE

Figure 2a reports the final regret on multi-class tasks. Numerical values are reported in Table 4 (Appendix C.1). `NeuralCBP` consistently achieves the lowest final regret on the four datasets considered. The improvement in final regret

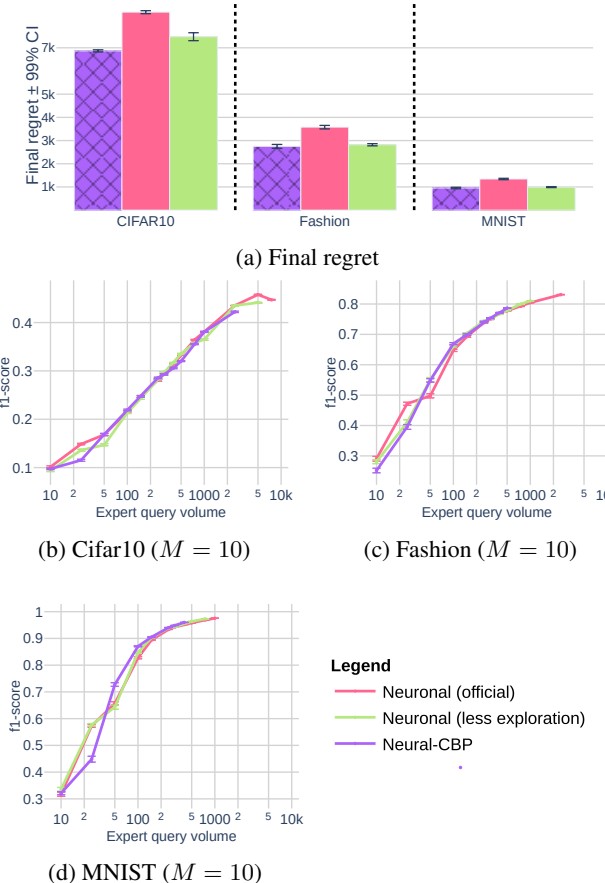

(a) Final regret

(b) Cifar10 ($M = 10$)

(c) Fashion ($M = 10$)

(d) MNIST ($M = 10$)

**Legend**

Neuronal (official)

Neuronal (less exploration)

Neural-CBP

Figure 3: Performance on multi-class OAL with LeNet.

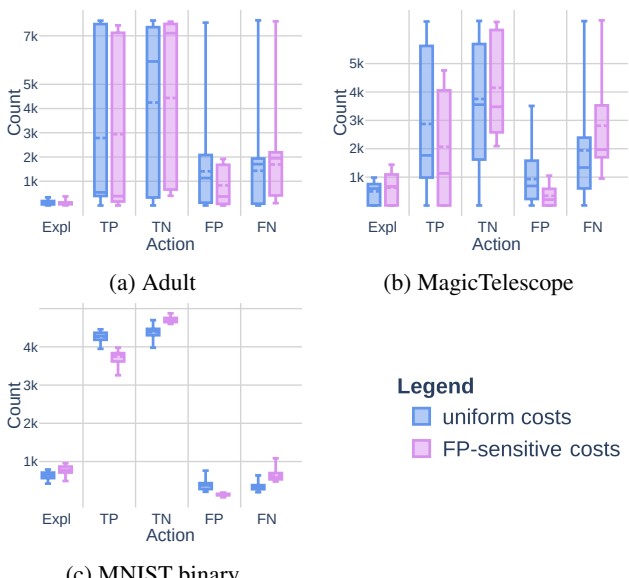

(a) Adult

(b) MagicTelescope

(c) MNIST binary

**Legend**

uniform costs

FP-sensitive costs

Figure 4: Distribution of the number of expert queries (Expl), true positives (TP), true negatives (TN), false positives (FP), and false negatives (FN) for `NeuralCBP` playing Label-efficient with uniform cost vs high cost on FP. Intrinsic goal of FP-sensitive: Minimize the count of FP.

performance is statistically significant for MNIST, Fashion and adult datasets (all p-values$< 0.01$), and `NeuralCBP` achieves identical performance to `Neuronal` (less exploration) on covertype. The improvement ranges from $15\%$ to $40\%$ over the second best baseline (`Neuronal` with less exploration) on MNIST, Fashion, and shuttle datasets.

Figures 2b, 2c, 2d and 2e show that given the same volume of expert queries, `NeuralCBP` achieves a comparable or better f1-score performance on the test sets. Other strategies may achieve a f1-score performance increase at the expense of significantly more expert queries. However, the generalization improvement obtained from these additional labelled observations is not reflected in terms of final regret performance.

**Robustness across different neural architectures.** We now evaluate `Neuronal` and `NeuralCBP` on a set of multi-class tasks using the convolutional architecture LeNet [LeCun et al., 1998]. `INeural` is omitted because it requires one-hot encodings of observations in the action space, which is not scalable when operating over multi-dimensional tensor observations. See numeric details in Table 6.

Figure 3a shows that `NeuralCBP` achieves the best final

regret performance. Furthermore, `NeuralCBP` achieves a f1-score performance comparable to `Neuronal` for equivalent volumes of expert queries. This is expected as the underlying networks $f_1$, $f_2$ and embeddings are the same for both approaches. Although `NeuralCBP` and `Neuronal` curves overlap along the f1-score axis (y-axis), it is worth noting that `NeuralCBP` has a smaller label complexity (x-axis). `Neuronal` (official) and `Neuronal` (less exploration) consume more expert queries, which is not translated into an improved final regret performance.

## 5.3 SPECIFYING A COST STRUCTURE

In this experiment, we investigate the impact of cost-sensitivity on the action selection strategy of `NeuralCBP`. To our knowledge, `NeuralCBP` is the only applicable strategy in cost-sensitive OAL tasks. We consider the previous binary OAL tasks conducted on Adult, MagicTelescope, and the modified MNIST datasets.

Recall from Section 3 that the original label efficient game corresponds to a binary classification task where the costs of prediction errors and expert queries are equal to 1 across the two classes. We refer to this formulation as the **uniform costs** case. We also consider a cost-sensitive variation where the cost of false negatives (FN) is twice as low as the cost of false positives (FP). We refer to this cost-sensitive variation as the **FP-sensitive costs** case. In the FP-sensitive case, the intrinsic goal is to minimize the amount of false positives. For example, this could refer to a learning sys-

tem constrained to minimize incorrect positive detection in medical screenings (HIV, cancer, etc).

Figure 4 illustrates the influence of the cost structure on `NeuralCBP`. We measure the count distribution (mean, median, 1st and 3rd quartiles) of expert queries, true positives (TP), true negatives (TN), false positives (FP), and false negatives (FP), over the 25 runs. In Adult and MagicTelescope datasets (Figures 4a and 4b), the third quartile of the FP count is approximately 3 times smaller under the FP-sensitive cost structure. On MNIST binary (Figure 4c), the mean FP count is $336 \pm 112.24$ (1-std) in the uniform case; this value drops to $136 \pm 30$ (1-std) in the FP-sensitive case. These numeric results show that `NeuralCBP` successfully accounts for the specified FP-sensitive cost structure.

## 6 DISCUSSION

**Partial monitoring feedback.** `NeuralCBP` is a strategy designed for the partial monitoring setting. To account for partial monitoring games, `NeuralCBP` operates a distinction between incurred costs (not learned, specified in the cost matrix **C**), observable feedbacks (not learned, specified in the feedback matrix **H**) and feedback distribution (learned components noted $\hat{\pi}_a(x_t)$ for each action $a$). The proposed PM-based formulation enables to specify a cost structure and possibly the presence of multiple experts.

**Distinct exploration principles.** In existing `EENets` strategies, the predictions of the network $f_2$ are added to the predictions of $f_1$ to compute an *upper confidence bound* on the predictions. Then, the magnitude of the difference between the top two predictions drives the exploration. In contrast, the predictions of network $f_2$ in `NeuralCBP` contribute to a *successive elimination* criteria (defined in Section 4.3) that checks whether the upper and lower confidence bounds of two different actions overlaps or not.

**Sensitivity to hyper-parameters** In `Neuronal` and `INeural` strategies, the decision of querying the expert is based on the difference between the top two class predictions. If the difference is greater than a slack term (obtained from the theory), the strategy asks for an expert label. The slack term provided by the theory is not usually computed in practice Ban et al. [2024]. Therefore, the user must select a proxy $\gamma$ of the slack term. As demonstrated in our experiments, we have evaluated the official ($\gamma = 0.6$) instances of `Neuronal` and `INeural`, as well as instances that induce less exploration ($\gamma = 3$). We observe from Figures 1, 2, and 3, that the final regret performance of `Neuronal` and `INeural` on a specific dataset is sensitive to an appropriate choice of $\gamma$. One benefit of `NeuralCBP` over `Neuronal` and `INeural` is that the exploration is driven by a successive elimination criteria that is hyper-parameter free. This is relevant in online learning where deployment

data is typically unknown in advance, making it difficult to tune hyper-parameters.

## 7 CONCLUSION

Our work demonstrates the potential of the partial monitoring framework in practice, a field traditionally supported by theoretical research. We leverage the PM framework to formulate OAL tasks and propose `NeuralCBP`, a PM strategy able to learn efficiently from neural networks. While the emphasis of this paper is on OAL, `NeuralCBP` is a general PM approach that can be applied to the broader diversity of partial monitoring games. Lastly, we demonstrate the empirical performance of `NeuralCBP` on a set of binary, multi-class and cost-sensitive OAL tasks, and highlight technical and empirical benefits over existing OAL strategies.

**Limitations** A limitation of `NeuralCBP` is that it does not scale well with large number of classes. Combinatorial PM strategies could address this limitation [Lin et al., 2014]. Furthermore, `NeuralCBP` does not capture the multi-expert case. The multi-expert case is studied in Dekel et al. [2012], Kumar et al. [2022] without the PM framework but a PM perspective based on Kirschner et al. [2020, 2023] could be an avenue of future research.

### Author Contributions

Maxime Heuillet: conceptualization, methodology, empirical investigation, visualizations, implementation, writing (original draft, editing), funding acquisition. Ola Ahmad: conceptualization, writing (review, editing), supervision. Audrey Durand: conceptualization, writing (review, editing), supervision, funding acquisition.

### Acknowledgements

This work was funded through a Mitacs Accelerate grant. We thank Alliance Canada and Calcul Quebec for access to computational resources and staff expertise consultation. We would like to thank Dr. Yikun Ban for answering our technical questions about `INeural` and `Neuronal`. We also acknowledge the library pmlib of Tanguy Urvoy that was helpful to implement `NeuralCBP` and PM game environments.

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

# Neural Active Learning Meets the Partial Monitoring Framework (Supplementary Material)

**Maxime Heuillet**[1]  **Ola Ahmad**[1,3]  **Audrey Durand**[1,2]

[1]Université Laval, Canada
[2]Canada-CIFAR AI Chair, Mila, Canada
[3]Thales Research and Technology (cortAIx), Canada,

| Notation | Definition | Observable by the agent? |
|---|---|:---:|
| $N$ | Number of actions | ✓ |
| $M$ | Number of outcomes | ✓ |
| $E$ | Number of experts | ✓ |
| $\Sigma$ | Feedback space (space of symbols) | ✓ |
| $\mathbf{C} \in [\mathbf{0,1}]^{\mathbf{N \times M}}$ | Cost matrix | ✓ |
| $\mathbf{c_i}$ | Row $i$ in matrix $\mathbf{C}$ (associated with action $i$) | ✓ |
| $\mathbf{H} \in \mathbf{\Sigma^{N \times M}}$ | Feedback matrix | ✓ |
| $\mathbf{h_i}$ | Row $i$ in matrix $\mathbf{H}$ (associated with action $i$) | ✓ |
| $\sigma_i$ | Number of unique feedback symbols induced by action $i$ (i.e. on row $i$ of $H$) | ✓ |
| $\Delta_M$ | Probability simplex of dimension $M$ (i.e. over the outcome space) | ✓ |
| $\Delta_{\sigma_i}$ | Probability simplex of dimension $\sigma_i$ (i.e. over the symbol space induced by action $i$) | ✓ |
| $T$ | Total number of rounds in a game (horizon) | ✗ |
| $i_t$ | Action played by the agent at round $t$ | ✓ |
| $y_t$ | Outcome at round $t$ | ✗ |
| $h_t$ | Feedback symbol at round $t$ | ✓ |
| $\mathbf{H[i_t, y_t]}$ | Element in matrix $\mathbf{H}$ at row $I_t$ and column $J_t$ (i.e. feedback received at round $t$) | ✓ |
| $\mathbf{C[i_t, y_t]}$ | Element in matrix $\mathbf{C}$ at row $I_t$ and column $J_t$ (i.e. loss incurred at round $t$) | ✗ |
| $\mathcal{X}$ | Observation space | ✗ |
| $\mathcal{I}$ | Set of informative actions | ✓ |
| $\Sigma_{\mathcal{I}}$ | Valid feedback symbols | ✓ |
| $\Pi(x_t)$ | Set of informative feedback distributions | ✗ |
| $x_t \in \mathcal{X}$ | Observation received at time $t$ | ✓ |
| $X_{i,t}$ | History of end-to-end embeddings for action $i$ up to time $t$ | ✓ |
| $G_{i,t}$ | Gram matrix for action $i$ up to time $t$ | ✓ |
| $p(x_t) \in \Delta_M$ | Outcome distribution | ✗ |
| $\mathcal{O}_i \subseteq \Delta_M$ | Cell of action $i$ | ✓ |
| $\Sigma_i$ | Enumeration of symbols sorted by order of appearance in $h_i$ | ✓ |
| $S_i \in \{0,1\}^{\sigma_i \times M}$ | Signal matrix of action $i$ | ✓ |
| $\pi_i(x_t) \in \Delta_{\sigma_i}$ | Distribution for the unique feedback symbols induced by action $i$ | ✗ |
| $\delta_{i,j}(x_t)$ | Expected loss difference between action $i$ and $j$ | ✗ |
| $\mathcal{P}$ | Set of Pareto optimal actions (i.e. set of actions) | ✓ |
| $\mathcal{N}$ | Set of neighbor action pairs (i.e. set of pairs of actions) | ✓ |
| $\mathcal{U}(t)$ | Set of confident action pairs (i.e. set of pairs of actions) | ✓ |
| $V_{i,j}$ | Observer set for pair $i, j$ (i.e. set of actions ) | ✓ |
| $v_{ija}$ | Observer vector associated with $V_{i,j}$ (index $a$ indicates to which action in $V_{i,j}$ it is associated to) | ✓ |
| $z_{i,j}(t)$ | Confidence for a pair $\{i, j\}$ at round $t$ | ✓ |
| $D(t) \subseteq \Delta_M$ | Sub-space of the simplex based on constraints in $\mathcal{U}(t)$, it includes $p^\star$ with high confidence | ✓ |
| $N_{i,j}^+$ | Neighbor action set for pair $i, j$ (set of actions) | ✓ |
| $\mathcal{P}(t)$ | Plausible subset of $\mathcal{P}$ given $D(t)$ (set of actions) | ✓ |
| $\mathcal{N}(t)$ | Plausible subset of $\mathcal{N}$ given $D(t)$ (set of pairs of actions) | ✓ |
| $\mathcal{R}(x_t)$ | Set of underplayed actions at time $t$ (set of actions) | ✓ |
| $e(\cdot)$ | One hot encoding | ✓ |
| $\mathcal{S}(t)$ | Final set of actions considered by CBP (set of actions) | ✓ |
| $W_a = \max_{\{i,j\} \in \mathcal{N}} \|v_{ija}\|_\infty$ | Weight of an action | ✓ |

Table 1: List of notations

## A ANALYSIS OF THE LABEL EFFICIENT GAME

The original label-efficient game [Helmbold et al., 1997] is defined by the following cost and feedback matrices:

$$
\mathbf{C} = \begin{array}{c} \text{pred. class A} \\ \text{pred. class B} \\ \text{expert} \end{array}
\begin{array}{cc} \text{class } A & \text{class } B \end{array}
\begin{bmatrix} 0 & 1 \\ 1 & 0 \\ 1 & 1 \end{bmatrix},
\mathbf{H} = \begin{array}{cc} \text{class } A & \text{class } B \end{array}
\begin{bmatrix} \Diamond & \Diamond \\ \wedge & \wedge \\ \perp & \odot \end{bmatrix}.
$$

The game includes a set of $N = 3$ possible actions and $M = 2$ possible outcomes (class A, and class B). For actions 1 and 2, there is $\sigma_1 = \sigma_2 = 1$ unique feedback symbol. For action 3, there is $\sigma_3 = 2$ feedback symbols, and the enumeration is $\{\perp, \odot\}$. Therefore, the set of informative actions is $\mathcal{I} = \{3\}$.

**Signal Matrices:** The dimension of the signal matrices are such that $S_1 \in \{0,1\}^{1 \times 2}$ and $S_2 \in \{0,1\}^{1 \times 2}$ and $S_3 \in \{0,1\}^{2 \times 2}$. The matrices verify:

$$
S_1 = \begin{bmatrix} 1 & 1 \end{bmatrix}, \quad S_2 = \begin{bmatrix} 1 & 1 \end{bmatrix}, \quad S_3 = \begin{bmatrix} 1 & 0 \\ 0 & 1 \end{bmatrix}
$$

The outcome distribution is noted $p^\star = [p_A, p_B]^\top$.

**Cells:** Each action can be associated to a sub-space of the probability simplex noted *cell* (see Definition 2.1):

- For action 1, we have: $\mathcal{O}_1 = \{p \in \Delta_M, \forall j \in \{1, \ldots, N\}, (\mathbf{c}_1 - \mathbf{c}_j)p \le 0\}$. This probability space corresponds to the following constraints:
$$
\begin{bmatrix} \mathbf{c}_1 - \mathbf{c}_1 \\ \mathbf{c}_1 - \mathbf{c}_2 \\ \mathbf{c}_1 - \mathbf{c}_3 \end{bmatrix} p = \begin{bmatrix} 0 & 0 \\ -1 & 1 \\ -1 & 0 \end{bmatrix} p \le 0
$$

  The first constraint $(\mathbf{c}_1 - \mathbf{c}_1)p \le 0$ is always verified. The second constraint $(\mathbf{c}_1 - \mathbf{c}_2)p \le 0$ implies $-p_A + p_B \le 0 \iff p_B \le p_A$. The third constraint $(\mathbf{c}_1 - \mathbf{c}_3)p \le 0$ implies $-p_A \le 0 \iff p_A \ge 0$.

- For action 2, we have: $\mathcal{O}_2 = \{p \in \Delta_M, \forall j \in \{1, \ldots, N\}, (\mathbf{c}_2 - \mathbf{c}_j)p \le 0\}$. This probability space corresponds to the following constraints:
$$
\begin{bmatrix} \mathbf{c}_2 - \mathbf{c}_1 \\ \mathbf{c}_2 - \mathbf{c}_2 \\ \mathbf{c}_2 - \mathbf{c}_3 \end{bmatrix} p = \begin{bmatrix} 1 & -1 \\ 0 & 0 \\ 0 & -1 \end{bmatrix} p \le 0
$$

  The second constraint $(\mathbf{c}_2 - \mathbf{c}_2)p \le 0$ is always satisfied. The first constraint $(\mathbf{c}_2 - \mathbf{c}_1)p \le 0$ implies $p_A - p_B \le 0 \iff p_A \le p_B$. The third constraint $(\mathbf{c}_2 - \mathbf{c}_1)p \le 0$ implies $-p_B \le 0 \iff p_B \ge 0$.

- For action 3, we have: $\mathcal{O}_3 = \{p \in \Delta_M, \forall j \in \{1, \ldots, N\}, (\mathbf{c}_3 - \mathbf{c}_j)p \le 0\}$. This probability space corresponds to the following constraints:
$$
\begin{bmatrix} \mathbf{c}_3 - \mathbf{c}_1 \\ \mathbf{c}_3 - \mathbf{c}_2 \\ \mathbf{c}_3 - \mathbf{c}_3 \end{bmatrix} p = \begin{bmatrix} 1 & 0 \\ 0 & 1 \\ 0 & 0 \end{bmatrix} p \le 0
$$

  The third constraint $(\mathbf{c}_1 - \mathbf{c}_1)p \le 0$ is always verified. The first constraint $(\mathbf{c}_1 - \mathbf{c}_2)p \le 0$ implies $p_A \le 0$ and the second constraint $(\mathbf{c}_1 - \mathbf{c}_3)p \le 0$ implies $p_B \le 0$. There exist no probability vector in $\Delta_M$ satisfying these three constraints at the same time.

**Pareto optimal actions:** From the analysis of the cells, we have $\mathcal{O}_3 = \emptyset$. Therefore, action 3 is dominated, according to Definition 2.1. The remaining actions 1 and 2 are Pareto optimal because their respective cells are not included in one another, i.e. $\mathcal{P} = \{1, 2\}$.

**Neighbor actions:** In this paragraph, we will determine whether action 1 and 2 are a neighbor pair.

$$\mathcal{O}_1 \cap \mathcal{O}_2 = \begin{cases} p_B \leq p_A \\ p_A \geq 0 \\ p_A \leq p_B \\ p_B \geq 0 \end{cases}$$

The only point in this vector space is $\begin{bmatrix} 0.5 & 0.5 \end{bmatrix}^\top$. Therefore, $\dim(\mathcal{O}_1 \cap \mathcal{O}_2) = 0 = M - 2$ and the pair $\{1, 2\}$ is a neighbor pair, i.e. $\mathcal{N} = \{\{1, 2\}, \}$.

**Neighbor action set:** This set is defined as $N_{ij}^+ = \{k \in \{1, \dots, N\}, \mathcal{O}_i \cap \mathcal{O}_j \subset \mathcal{O}_k\}$. This yields: $N_{1,2}^+ = N_{2,1}^+ = [1, 2]$ because the cell of action 3 is empty.

**Informative action set** Action 3 is the only informative action because $\sigma_3 = 2 \geq 1$.

**Observer set:** We have: $V_{1,2} = \{3\}$ same applies to $V_{2,1} = \{3\}$, because action 3 is the only informative action.

**Observer vector:** For the pair $\{1, 2\}$, we have to find $v_{ija}, a \in V_{ij}$ such that $C_1^\top - C_2^\top = \sum_{a \in V_{ij}} S_i^T v_{ija}$, according to Definition 4.3. Choosing and $v_{121}^\top = \begin{bmatrix} -1 & 1 \end{bmatrix}$ verifies the relation:

$$\mathbf{c}_1^\top - \mathbf{c}_2^\top = \begin{bmatrix} -1 \\ 1 \end{bmatrix} = \begin{bmatrix} 1 & 0 \\ 0 & 1 \end{bmatrix} \begin{bmatrix} -1 \\ 1 \end{bmatrix}$$

# B  IMPLEMENTATION DETAILS

The pseudo-code in Algorithm 2 details how the `EENets` is updated.

---

**Algorithm 2:** Update EENet with gradient descent

**input** : $\theta_1, \theta_2$

Epoch number $K_1$ and $K_2$, learning rate $\eta_1$ and $\eta_2$

Initialize $\theta_1^{(0)} = \theta_1$

**for** $k \in \{1, \dots, K_1\}$ **do**

$\quad \left\lfloor \ \theta_1^{(k)} = \theta_1^{(k-1)} - \mu_1 \nabla_{\theta_1^{(k-1)}} \mathcal{L}_1(\theta_1^{(k-1)}) \right. ;$

$\tilde{\theta}_1 = \theta_1^{(K_1)}$

Initialize $\theta_2^{(0)} = \theta_2$

**for** $k \in \{1, \dots, K_2\}$ **do**

$\quad \left\lfloor \ \theta_2^{(k)} = \theta_2^{(k-1)} - \mu_2 \nabla_{\theta_2^{(k-1)}} \mathcal{L}_2(\theta_2^{(k-1)}) \right. ;$

$\tilde{\theta}_2 = \theta_2^{(K_2)}$

**output** : $\tilde{\theta}_1, \tilde{\theta}_2$

---

# C  EXPERIMENT DETAILS.

The neural components of the strategies refer to two networks $f_1$ and $f_2$ for `NeuralCBP`, `Neuronal` and `INeural`; $f_1$ is trained using the MSE loss functions $\mathcal{L}_1$ and $f_2$ with $\mathcal{L}_2$. For strategies `Cesa` and `Margin`, the neural component corresponds to one network $f_1$, which is trained using the MSE loss function $\mathcal{L}_1$. In the experiments with a MLP, $f_1$ is a MLP architecture of width $m = 100$ and depth $L = 2$, and $f_2$ is a MLP of width $m = 100$ and depth $L = 2$. In the experiments with a LeNet, $f_1$ is a LeNet architecture LeCun et al. [1998], and $f_2$ is a MLP of width $m = 100$ and depth $L = 2$.

**Update protocol for the neural components of the strategies.** At the beginning of the game, each strategy plays each action once. Then, to save compute, we perform updates at every round for the first $N \leq t \leq 50$ steps. We update every 50 rounds for $t \leq 1000$. Finally, we update every 500 rounds when $t \geq 1000$. An equivalent update protocol has been used in related neural online learning literature Xu et al. [2022]. This update protocol is implemented for all the strategies considered in the experiments.

**End-to-end embedding down sampling.** Both `Neuronal` and `NeuralCBP` use the *end-to-end embedding* (see Definition 4.8). Due to the dimension of a flattened gradient, the embedding received as input to $f_2$ requires a down-sampling. Similarly to Ban et al. [2024], we use a *block-reduction averaging operator*. When $f_1$ is based on a MLP architecture, the reduction parameter to 51 following Ban et al. [2024]. When $f_1$ is based on a LeNet architecture, we set the reduction parameter to 51 for MNIST and FASHION datasets. For CIFAR10, we increase the block averaging to $153 = 51 \times 3$ to account for the three color channels (RGB) of CIFAR10 observations.

**`NeuralCBP`.** To speed up compute, the inversion and updates of the Gram matrix are performed on GPU, using the Sherman-Morison update [Sherman et al., 1950]. We set $f(t) = \alpha^{1/3} t^{2/3} \log(t)^{1/3}$, $\eta_a = W_a^{2/3}$ and $\alpha = 1.01$ according to previous literature Heuillet et al. [2024]. This combination of parameters is justified by the theoretical analysis of `CBP` and is not meant to be tuned further.

To update $f_1$ and $f_2$, we use the Adam optimizer, with the learning rate set to the default value $\mu_1 = \mu_2 = 0.001$ (both for MLP and LeNet architectures). Following Ban et al. [2024], we set the batch size to 64 and the number of epochs to 40. We performed a grid search for the learning rate over $\{0.0001, 0.001\}$ and found that the value $0.001$ performs best.

**`Neuronal`** The strategy `Neuronal` admits a hyper-parameter $\gamma$ that influences the amount of exploration. For `Neuronal` (official), we set $\gamma = 6$, as reported in Ban et al. [2024]. We also consider the instance $\gamma = 3$ for `Neuronal` (less exploration), which exhibits less exploration.

Following Ban et al. [2024], we set the batch size to 64 and the number of epochs to $K_1 = K_2 = 40$. For $f_1$ and $f_2$, we use the Adam optimizer, with the value for the learning rate set at $\mu_1 = \mu_2 = 0.001$ for both the MLP, and LeNet architecture.

We performed a grid search and report empirical findings for the learning rate over $\{0.0001, 0.001\}$. We observed from Tables 2, 3, 4, 5 and 6, 7 that a learning rate $\mu_1 = \mu_2 = 0.001$ performs best on most datasets for both instances of `Neuronal` (official and $\gamma = 3$).

**`INeural`** The exploration parameter is set to $\gamma = 6$ for `INeural` (official) and to $\gamma = 3$ for `INeural` (less exploration). Note that the networks $f_1$ and $f_2$ of `INeural` have a different input dimension, as they require input observations to be one-hot-encoded over the action space.

For $f_1$ and $f_2$, we use the Adam optimizer, with the default value for the learning rate set at $\mu_1 = \mu_2 = 0.001$. The batch size is set to 64 and the number of epochs to 40. This approach is known to be outperformed by `Neuronal`, we used the set of optimal hyper-parameters reported in Ban et al. [2024].

**`Cesa`** The exploration strategy of the approach `Cesa` is hyper-parameter free. For the network $f_1$, we use the Adam optimizer, with the default value for the learning rate set at $\mu_1 = 0.001$. The batch size is set to 64 and the number of epochs to $K_1 = 40$.

**`Margin`** The exploration parameter of the `Margin` approach is set to 1. For network $f_1$, we use the Adam optimizer, with the default value for the learning rate set at $\mu_1 = 0.001$. The batch size is set to 64 and the number of epochs to $K_1 = 40$.

## C.1 NUMERICAL RESULTS

In this Appendix, we report the empirical performance of `Neuronal` with a learning rate set to $\mu_1 = \mu_2 = 0.0001$. We also report numeric details for all the figures reported in the main body and appendix.

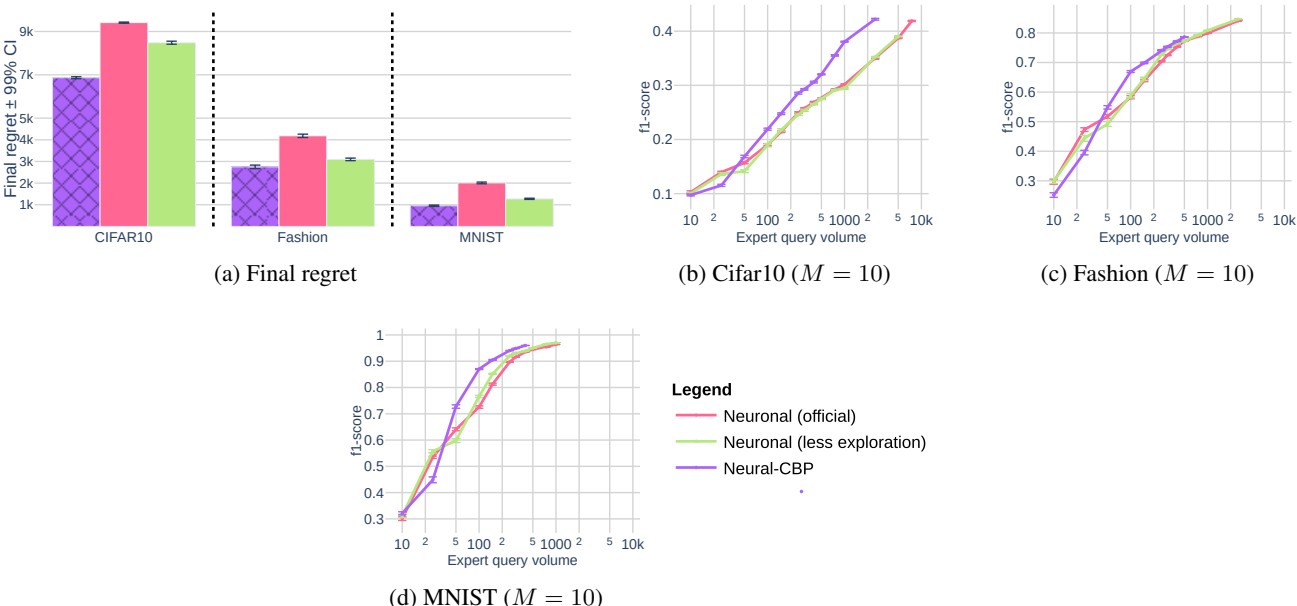

(a) Final regret (lower is better)

(a) Final regret

(b) Adult

(c) Magic Telescope

(b) Fashion ($M = 10$)

(c) MNIST ($M = 10$)

(d) MNIST Binary

**Legend**
- IneurAL (official)
- IneurAL (less exploration)
- Neuronal (official)
- Neuronal (less exploration)
- Margin
- Cesa
- Neural-CBP

(d) Covertype ($M = 7$)

(e) Shuttle ($M = 7$)

- IneurAL (official)
- IneurAL (less exploration)
- Neuronal (official)
- Neuronal (less exploration)
- Neural-CBP

Figure 5: Performance on binary OAL with MLP. `Neuronal` with $\mu_1 = \mu_2 = 0.0001$.

Figure 6: Performance on multi-class OAL with MLP. `Neuronal` with $\mu_1 = \mu_2 = 0.0001$.

(a) Final regret

(b) Cifar10 ($M = 10$)

(c) Fashion ($M = 10$)

**Legend**
- Neuronal (official)
- Neuronal (less exploration)
- Neural-CBP

(d) MNIST ($M = 10$)

Figure 7: Performance on multi-class OAL with LeNet. `Neuronal` with $\mu_1 = \mu_2 = 0.0001$.

| Dataset | Approach | Mean regret | p-value | win count | Mean Exploration | p-value (exploration) |
|---|---|---|---|---|---|---|
| MNIST binary | Neural-CBP | 1351.92 | 1.0 | 10.0 | 638.32 | 1.0 |
| MNIST binary | IneurAL (official) | 1701.72 | 0.0 | 0.0 | 1311.44 | 0.0 |
| MNIST binary | IneurAL (less exploration) | 1701.28 | 0.103 | 2.0 | 624.96 | 0.756 |
| MNIST binary | Neuronal (official) | 1566.16 | 0.0 | 0.0 | 1363.88 | 0.0 |
| MNIST binary | Neuronal (less exploration) | 1646.16 | 0.172 | 13.0 | 778.4 | 0.01 |
| MNIST binary | Cesa | 2627.52 | 0.0 | 0.0 | 303.28 | 0.0 |
| MNIST binary | Margin | 3169.6 | 0.0 | 0.0 | 100.28 | 0.0 |
| MagicTelescope | Neural-CBP | 3370.36 | 1.0 | 14.0 | 496.68 | 1.0 |
| MagicTelescope | IneurAL (official) | 7343.04 | 0.0 | 0.0 | 6452.8 | 0.0 |
| MagicTelescope | IneurAL (less exploration) | 4473.2 | 0.0 | 0.0 | 3138.28 | 0.0 |
| MagicTelescope | Neuronal (official) | 3716.8 | 0.207 | 0.0 | 1604.84 | 0.0 |
| MagicTelescope | Neuronal (less exploration) | 3287.0 | 0.769 | 7.0 | 444.64 | 0.613 |
| MagicTelescope | Cesa | 3245.64 | 0.623 | 1.0 | 328.76 | 0.033 |
| MagicTelescope | Margin | 3574.12 | 0.483 | 3.0 | 58.04 | 0.0 |
| adult | Neural-CBP | 2968.88 | 1.0 | 10.0 | 127.64 | 1.0 |
| adult | IneurAL (official) | 8243.12 | 0.0 | 0.0 | 7610.2 | 0.0 |
| adult | IneurAL (less exploration) | 3909.48 | 0.027 | 0.0 | 1136.24 | 0.0 |
| adult | Neuronal (official) | 3222.24 | 0.551 | 1.0 | 951.72 | 0.025 |
| adult | Neuronal (less exploration) | 3035.92 | 0.884 | 5.0 | 89.64 | 0.379 |
| adult | Cesa | 3180.0 | 0.643 | 3.0 | 320.12 | 0.001 |
| adult | Margin | 3166.56 | 0.682 | 6.0 | 19.92 | 0.0 |

Table 2: Supplement for Section 5.1 presented in the main paper (see Figure 1). `Neuronal` with $\mu_1 = \mu_2 = 0.001$. Mean regret: average regret at the last step ($T = 10k$). P-value: Welch's t-test on the distribution of regrets at the last step, with `NeuralCBP` as reference (p-value $> 0.05$ means no statistical difference). Win count: number of times a given strategy achieved the lowest final regret (ties included). Mean exploration: average number of expert-verified observations. P-value (exploration): Welch's t-test on the distribution of number of expert queries.

| Dataset | Approach | Mean regret | p-value | win count | Mean Exploration | p-value (exploration) |
|---|---|---|---|---|---|---|
| MNIST binary | Neural-CBP | 1351.92 | 1.0 | 15.0 | 638.32 | 1.0 |
| MNIST binary | IneurAL (official) | 1701.72 | 0.0 | 0.0 | 1311.44 | 0.0 |
| MNIST binary | IneurAL (less exploration) | 1701.28 | 0.103 | 8.0 | 624.96 | 0.756 |
| MNIST binary | Neuronal (official) | 1730.08 | 0.0 | 0.0 | 1558.76 | 0.0 |
| MNIST binary | Neuronal (less exploration) | 1440.48 | 0.005 | 2.0 | 987.84 | 0.0 |
| MNIST binary | Cesa | 2627.52 | 0.0 | 0.0 | 303.28 | 0.0 |
| MNIST binary | Margin | 3169.6 | 0.0 | 0.0 | 100.28 | 0.0 |
| MagicTelescope | Neural-CBP | 3370.36 | 1.0 | 16.0 | 496.68 | 1.0 |
| MagicTelescope | IneurAL (official) | 7343.04 | 0.0 | 0.0 | 6452.8 | 0.0 |
| MagicTelescope | IneurAL (less exploration) | 4473.2 | 0.0 | 0.0 | 3138.28 | 0.0 |
| MagicTelescope | Neuronal (official) | 4226.96 | 0.002 | 0.0 | 2555.96 | 0.0 |
| MagicTelescope | Neuronal (less exploration) | 3372.2 | 0.994 | 4.0 | 556.52 | 0.631 |
| MagicTelescope | Cesa | 3245.64 | 0.623 | 3.0 | 328.76 | 0.033 |
| MagicTelescope | Margin | 3574.12 | 0.483 | 2.0 | 58.04 | 0.0 |
| adult | Neural-CBP | 2968.88 | 1.0 | 10.0 | 127.64 | 1.0 |
| adult | IneurAL (official) | 8243.12 | 0.0 | 0.0 | 7610.2 | 0.0 |
| adult | IneurAL (less exploration) | 3909.48 | 0.027 | 2.0 | 1136.24 | 0.0 |
| adult | Neuronal (official) | 4819.44 | 0.001 | 1.0 | 3395.52 | 0.0 |
| adult | Neuronal (less exploration) | 2523.64 | 0.155 | 3.0 | 305.96 | 0.032 |
| adult | Cesa | 3180.0 | 0.643 | 3.0 | 320.12 | 0.001 |
| adult | Margin | 3166.56 | 0.682 | 6.0 | 19.92 | 0.0 |

Table 3: Supplement for Figure 5. `Neuronal` with $\mu_1 = \mu_2 = 0.0001$. Mean regret: average regret at the last step ($T = 10k$). P-value: Welch's t-test on the distribution of regrets at the last step, with `NeuralCBP` as reference (p-value $> 0.05$ means no statistical difference). Win count: number of times a given strategy achieved the lowest final regret (ties included). Mean exploration: average number of expert-verified observations. P-value (exploration): Welch's t-test on the distribution of number of expert queries.

| Dataset | Approach | Mean regret | p-value | win count | Mean Exploration | p-value (exploration) |
|---|---|---|---|---|---|---|
| MNIST | Neural-CBP | 1811.96 | 1.0 | 25.0 | 1305.24 | 1.0 |
| MNIST | IneurAL (official) | 4016.84 | 0.0 | 0.0 | 3870.12 | 0.0 |
| MNIST | IneurAL (less exploration) | 2371.36 | 0.0 | 0.0 | 1818.56 | 0.0 |
| MNIST | Neuronal (official) | 3275.0 | 0.0 | 0.0 | 3224.84 | 0.0 |
| MNIST | Neuronal (less exploration) | 2213.16 | 0.0 | 0.0 | 2030.44 | 0.0 |
| Fashion | Neural-CBP | 2898.24 | 1.0 | 25.0 | 1523.92 | 1.0 |
| Fashion | IneurAL (official) | 4882.48 | 0.0 | 0.0 | 4382.88 | 0.0 |
| Fashion | IneurAL (less exploration) | 3437.76 | 0.0 | 0.0 | 2210.2 | 0.0 |
| Fashion | Neuronal (official) | 4466.76 | 0.0 | 0.0 | 4193.28 | 0.0 |
| Fashion | Neuronal (less exploration) | 3417.44 | 0.0 | 0.0 | 2777.44 | 0.0 |
| covertype | Neural-CBP | 5060.24 | 1.0 | 12.0 | 446.96 | 1.0 |
| covertype | IneurAL (official) | 9129.68 | 0.0 | 0.0 | 8235.72 | 0.0 |
| covertype | IneurAL (less exploration) | 7707.72 | 0.0 | 0.0 | 4862.12 | 0.0 |
| covertype | Neuronal (official) | 5970.96 | 0.0 | 0.0 | 4802.48 | 0.0 |
| covertype | Neuronal (less exploration) | 5060.92 | 0.997 | 13.0 | 1330.4 | 0.001 |
| shuttle | Neural-CBP | 907.56 | 1.0 | 14.0 | 190.08 | 1.0 |
| shuttle | IneurAL (official) | 7989.88 | 0.0 | 0.0 | 7901.0 | 0.0 |
| shuttle | IneurAL (less exploration) | 2810.08 | 0.0 | 1.0 | 2292.48 | 0.0 |
| shuttle | Neuronal (official) | 1547.08 | 0.004 | 7.0 | 183.48 | 0.902 |
| shuttle | Neuronal (less exploration) | 1524.0 | 0.003 | 4.0 | 157.16 | 0.437 |

Table 4: Numeric values in support of Section 5.2 presented in the main paper (see Figure 2). `Neuronal` with $\mu_1 = \mu_2 = 0.001$. Mean regret: average regret at the last step ($T = 10k$). P-value: Welch's t-test on the distribution of regrets at the last step, with `NeuralCBP` as reference (p-value $> 0.05$ means no statistical difference). Win count: number of times a given strategy achieved the lowest final regret (ties included). Mean exploration: average number of expert-verified observations. P-value (exploration): Welch's t-test on the distribution of number of expert queries.

| Dataset | Approach | Mean regret | p-value | win count | Mean Exploration | p-value (exploration) |
|---|---|---|---|---|---|---|
| MNIST | Neural-CBP | 1811.96 | 1.0 | 25.0 | 1305.24 | 1.0 |
| MNIST | IneurAL (official) | 4016.84 | 0.0 | 0.0 | 3870.12 | 0.0 |
| MNIST | IneurAL (less exploration) | 2371.36 | 0.0 | 0.0 | 1818.56 | 0.0 |
| MNIST | Neuronal (official) | 4123.8 | 0.0 | 0.0 | 4093.68 | 0.0 |
| MNIST | Neuronal (less exploration) | 2722.72 | 0.0 | 0.0 | 2605.16 | 0.0 |
| Fashion | Neural-CBP | 2898.24 | 1.0 | 25.0 | 1523.92 | 1.0 |
| Fashion | IneurAL (official) | 4882.48 | 0.0 | 0.0 | 4382.88 | 0.0 |
| Fashion | IneurAL (less exploration) | 3437.76 | 0.0 | 0.0 | 2210.2 | 0.0 |
| Fashion | Neuronal (official) | 4881.16 | 0.0 | 0.0 | 4709.56 | 0.0 |
| Fashion | Neuronal (less exploration) | 3584.28 | 0.0 | 0.0 | 3089.64 | 0.0 |
| covertype | Neural-CBP | 5060.24 | 1.0 | 20.0 | 446.96 | 1.0 |
| covertype | IneurAL (official) | 9129.68 | 0.0 | 1.0 | 8235.72 | 0.0 |
| covertype | IneurAL (less exploration) | 7707.72 | 0.0 | 0.0 | 4862.12 | 0.0 |
| covertype | Neuronal (official) | 7831.48 | 0.0 | 0.0 | 7287.32 | 0.0 |
| covertype | Neuronal (less exploration) | 5659.96 | 0.0 | 4.0 | 3431.8 | 0.0 |
| shuttle | Neural-CBP | 907.56 | 1.0 | 20.0 | 190.08 | 1.0 |
| shuttle | IneurAL (official) | 7989.88 | 0.0 | 0.0 | 7901.0 | 0.0 |
| shuttle | IneurAL (less exploration) | 2810.08 | 0.0 | 0.0 | 2292.48 | 0.0 |
| shuttle | Neuronal (official) | 1661.08 | 0.0 | 1.0 | 1277.04 | 0.0 |
| shuttle | Neuronal (less exploration) | 1666.0 | 0.0 | 4.0 | 404.56 | 0.011 |

Table 5: Numeric values in support of Figure 6. `Neuronal` with $\mu_1 = \mu_2 = 0.0001$. Mean regret: average regret at the last step ($T = 10k$). P-value: Welch's t-test on the distribution of regrets at the last step, with `NeuralCBP` as reference (p-value $> 0.05$ means no statistical difference). Win count: number of times a given strategy achieved the lowest final regret (ties included). Mean exploration: average number of expert-verified observations. P-value (exploration): Welch's t-test on the distribution of number of expert queries.

| Dataset | Approach | Mean regret | p-value | win count | Mean Exploration | p-value (exploration) |
|---------|----------|-------------|---------|-----------|------------------|----------------------|
| MNIST | Neural-CBP | 954.458 | 1.0 | 17.0 | 439.75 | 1.0 |
| MNIST | Neuronal (official) | 1342.24 | 0.0 | 0.0 | 1284.16 | 0.0 |
| MNIST | Neuronal (less exploration) | 992.6 | 0.015 | 7.0 | 817.92 | 0.0 |
| Fashion | Neural-CBP | 2749.72 | 1.0 | 20.0 | 606.52 | 1.0 |
| Fashion | Neuronal (official) | 3576.76 | 0.0 | 0.0 | 3027.96 | 0.0 |
| Fashion | Neuronal (less exploration) | 2818.8 | 0.072 | 6.0 | 1698.16 | 0.0 |
| CIFAR10 | Neural-CBP | 6865.92 | 1.0 | 25.0 | 2515.04 | 1.0 |
| CIFAR10 | Neuronal (official) | 8535.76 | 0.0 | 0.0 | 7519.36 | 0.0 |
| CIFAR10 | Neuronal (less exploration) | 7478.12 | 0.0 | 0.0 | 5114.48 | 0.0 |

Table 6: Numeric values in support of Section 5.2 presented in the main paper (see Figure 3). `Neuronal` with $\mu_1 = \mu_2 = 0.001$. Mean regret: average regret at the last step ($T = 10k$). P-value: Welch's t-test on the distribution of regrets at the last step, with `NeuralCBP` as reference (p-value $> 0.05$ means no statistical difference). Win count: number of times a given strategy achieved the lowest final regret (ties included). Mean exploration: average number of expert-verified observations. P-value (exploration): Welch's t-test on the distribution of number of expert queries.

| Dataset | Approach | Mean regret | p-value | win count | Mean Exploration | p-value (exploration) |
|---------|----------|-------------|---------|-----------|------------------|----------------------|
| MNIST | Neural-CBP | 954.458 | 1.0 | 24.0 | 439.75 | 1.0 |
| MNIST | Neuronal (official) | 2009.04 | 0.0 | 0.0 | 1975.8 | 0.0 |
| MNIST | Neuronal (less exploration) | 1274.24 | 0.0 | 0.0 | 1154.96 | 0.0 |
| Fashion | Neural-CBP | 2749.72 | 1.0 | 24.0 | 606.52 | 1.0 |
| Fashion | Neuronal (official) | 4182.96 | 0.0 | 0.0 | 3919.52 | 0.0 |
| Fashion | Neuronal (less exploration) | 3095.28 | 0.0 | 1.0 | 2352.64 | 0.0 |
| CIFAR10 | Neural-CBP | 6865.92 | 1.0 | 25.0 | 2515.04 | 1.0 |
| CIFAR10 | Neuronal (official) | 9410.12 | 0.0 | 0.0 | 9130.44 | 0.0 |
| CIFAR10 | Neuronal (less exploration) | 8483.68 | 0.0 | 0.0 | 6674.24 | 0.0 |

Table 7: Numeric values in support of Figure 7. `Neuronal` with $\mu_1 = \mu_2 = 0.0001$. Mean regret: average regret at the last step ($T = 10k$). P-value: Welch's t-test on the distribution of regrets at the last step, with `NeuralCBP` as reference (p-value $> 0.05$ means no statistical difference). Win count: number of times a given strategy achieved the lowest final regret (ties included). Mean exploration: average number of expert-verified observations. P-value (exploration): Welch's t-test on the distribution of number of expert queries.