# OpenReview forum: "Neural Active Learning Meets the Partial Monitoring Framework"
_auai.org/UAI/2024/Conference — UAI 2024 poster_

### Official Review · Reviewer_RDCb · 2024-02-27

**Q2-1 Originality-Novelty:** 3
**Q2-2 Correctness-Technical Quality:** 3
**Q2-5 Clarity Of Writing:** 3

**Q1 Summary And Contributions:**

This paper focuses on the online-based active learning (OAL) setting, where an agent operates over a stream of observations and trades-off between the costly acquisition of information (labelled observations) and the cost of prediction errors. The authors propose a novel foundation for OAL tasks based on partial monitoring, a theoretical framework specialized in online learning from partially informative actions. The authors show that previously studied binary and multi-class OAL tasks are instances of partial monitoring. The authors expand the real-world potential of OAL by introducing a new class of cost-sensitive OAL tasks. Furthermore, the authors propose NeuralCBP, the first partial monitoring (PM) strategy that accounts for predictive uncertainty with deep neural networks. The empirical evaluations on open source datasets show that NeuralCBP is competitive with, and even outperforms state-of-the-art baselines on multiple binary, multi-class and cost-sensitive OAL tasks.

**Q2-3 Extent To Which Claims Are Supported By Evidence:**

2: Fair: the main claims are somewhat supported by evidence (but the experimental evaluation may be weak, or does not match entirely with the claims, important baselines may be missing, proofs contain important ideas but lack rigor, algorithmic details are only discussed superficially, references are imprecise, assumptions are not sufficiently motivated or explicated, etc.).

**Q2-4 Reproducibility:**

3: Good: key resources (e.g. proofs, code, data) are available and key details (e.g. proofs, experimental setup) are sufficiently well-described for competent researchers to confidently reproduce the main results.

**Q3 Main Strengths:**

1.	The studied problem, online-based active learning (OAL) in the partial monitoring setting, is interesting to the online learning literature, and can inspire potential future works.
2.	The authors design a strategy called NeuralCBP, which accounts for predictive uncertainty with deep neural networks.
3.	The authors present extensive experimental results to show that NeuralCBP is competitive with, and even outperforms state-of-the-art baselines on multiple binary, multi-class and cost-sensitive OAL tasks.

**Q4 Main Weakness:**

1.	It seems that the proposed NeuralCBP algorithm is a purely heuristic algorithm. Can the authors provide any theoretical guarantee for this algorithm, e.g., the regret bound. This is my main concern.
2.	For the baseline in experiments, can the authors compare to more existing online partial monitoring algorithms or their adaptations to this setting?

**Q5 Detailed Comments To The Authors:**

Please see the weaknesses above.

**Q9 Complying With Reviewing Instructions:**

Yes

---

> ### Author Rebuttal · Authors · 2024-04-05
>
> Thank you very much for your detailed and constructive comments, we are glad to respond to your questions.
>
> **Theoretical analysis of NeuralCBP.**
>
> Obtaining guarantees for NeuralCBP should be attainable, but it is unrealistic to get the analysis ready and correct in the time allowed by the rebuttal. Regret upper bounds for CBP methods in the linear and logistic setting have been developed [1,2]. Furthermore, NeuralCBP employs the Exploration Exploitation networks (EE-nets), which have been studied with the neural kernel tangent theorem [8].
>
> Obtaining theoretical guarantees for partial monitoring strategies in online active learning is an interesting avenue of future work. Traditionally, the online active learning community seeks distinct upper bounds on the generalization error and on the label complexity. However, partial monitoring strategies are natively compatible with guarantees in the form of a regret upper-bound. Although a regret upper bound would be interesting to the partial monitoring community, it would hardly compare with existing online active learning guarantees. This would be exciting for further research and improvement in future.
>
> The scope of our work is to establish a non-trivial connection between the partial monitoring framework and the active learning community. Our empirical results show that this connection is promising, as demonstrated by the competitive empirical performance of NeuralCBP. The nature of our contribution calls for convincing empirical evidence to make an interesting proof of concept. For example, the first neural approaches that were developed in the bandits setting were not accompanied with theoretical guarantees [9].
>
> **Choice of baselines in the experiments.**
>
> We would like to highlight that particular efforts have been made to include an extensive set of baselines in the experiments. For example, our experiments include [6], which is the current state-of-the-art to be presented at ICLR 2024 and that we outperform.
>
> In the binary case, we include baselines Cesa [3] and Margin [4], that we adapt to the neural setting. Cesa [3] and Margin [4] are well known baselines for the binary label efficient game with uniform cost structure [2]. Cesa [3] and Margin [4] are typically omitted from recent foundational works foundational in neural active learning [5, 6, 7]. Regarding pure partial monitoring baselines, the only remaining applicable strategies are restricted to the linear and logistic setting [1,2] and are based on CBP. We excluded them because our experiments require the ability to learn from a neural representation.
>
> In the cost-sensitive online active learning task (Section 5.3), NeuralCBP is the only applicable strategy. In these experiments, we focus on 10 class tasks while Cesa [3] and Margin [4] are binary strategies. Furthermore, Cesa [3] and Margin [4] require a uniform cost structure whereas the narrative of these experiments is to characterize the influence of a non-uniform cost structure.
>
>
> **References**
>
> [1] G. Bartok et al. “Partial monitoring with side information” CoLT (2014)
>
> [2] M. Heuillet et al. “Randomized Confidence Bounds in Stochastic Partial Monitoring” (arxiv 2024)
>
> [3] N. Cesa-Bianchi et al. “Worst-case analysis of selective sampling for linear classification”. JMLR (2016)
>
> [4] David Sculley et al. “Practical learning from one-sided feedback.” SIGKDD (2007)
>
> [5] A. Saran et. al “Streaming Active Learning with deep neural networks” ICML (2023)
>
> [6] Y. Ban et al. “Neural Active Learning Beyond Bandits” ICLR (2024)
>
> [7] De Salvo “Online Active Learning with Surrogate Loss Functions” Neurips (2021)
>
> [8] A. Jacot et al. “Neural Tangent Kernel: Convergence and Generalization in Neural Networks” Neurips (2018)
>
> [9] C. Riquelme et al. “Deep Bayesian Bandit Showdown” ICLR (2018)

---

### Official Review · Reviewer_iPL3 · 2024-03-22

**Q2-1 Originality-Novelty:** 3
**Q2-2 Correctness-Technical Quality:** 3
**Q2-5 Clarity Of Writing:** 2

**Q1 Summary And Contributions:**

This paper leverages the partial monitoring (PM) framework to model online-based active learning (OAL) tasks. In PM, an agent interacts with an environment through actions and receives limited feedback without direct observation of the environment's state. So that the framework helps in understanding and optimizing the trade-offs between exploring and exploiting. This paper then introduces NeuralCBP, a novel strategy within the PM framework that uses DNNs to estimate predictive uncertainty. This strategy is designed to work in scenarios where outcomes are not directly observable, and it helps in making informed decisions about when to query an expert for more information.

**Q2-3 Extent To Which Claims Are Supported By Evidence:**

3: Good: the main claims are supported by convincing evidence (in the form of adequate experimental evaluation, proofs, (pseudo-)code, references, assumptions).

**Q2-4 Reproducibility:**

4: Excellent: key resources (e.g. proofs, code, data) are available and key details (e.g. proof sketches, experimental setup) are comprehensively described for competent researchers to confidently and easily reproduce the main results.

**Q3 Main Strengths:**

1. This work integrates the theoretical framework of partial monitoring with practical deep-learning approaches, which let NeuralCBP be able to handle complex, high-dimensional data.

2. cost-sensitive OAL task consideration is a novel point since this was less explored previously.

**Q4 Main Weakness:**

NeuralCBP did not outperform many baselines in earlier stages of binary classification tasks.

**Q5 Detailed Comments To The Authors:**

Relying solely on the F1 score to evaluate classification tasks can be misleading, especially in cases of imbalanced datasets. Although most datasets used in the experiment are balanced datasets.

**Q9 Complying With Reviewing Instructions:**

Yes

---

> ### Author Rebuttal · Authors · 2024-04-05
>
> Thank you very much for your detailed and constructive comments, we are glad to respond to your questions.
>
> **Empirical performance of NeuralCBP in early stages of binary tasks.**
>
> To characterize the performance of NeuralCBP, we employ the f1-score and the regret. The f1-score is evaluated on a separate test dataset that is kept apart, while the regret is evaluated on the actual (online) stream observed by the agent.
>
> You are right to observe that the f1-score of NeuralCBP is outperformed by other baselines in the isolated case of the early stages of the binary adult dataset. However, the point of online active learning is to deploy an agent that can explore automatically and converge to an optimal solution without having to set a budget. In Figure 1.b, you can observe that NeuralCBP is one of the fastest strategies to stop exploring and when it does, it reaches a competitive f1-score performance.
>
> Furthermore, we would like to point out that the binary setting is the most competitive setting to benchmark NeuralCBP. The binary OAL setting has been the most studied in the literature, resulting in specialized baselines that are close to optimality (e.g. baselines Cesa [1] and Margin [2]). On the other end, NeuralCBP is a generic and more general approach that can be used to a wider variety of problems including multi-class problems with uniform or non-uniform costs.
>
> **Metrics to characterize performance.**
>
> Thank you for paying attention to this technical aspect. Our submitted manuscript reports the weighted f1-score based on class imbalance [3], which is different from the vanilla f1-score. We have edited the manuscript (page 6) to clarify this: “To evaluate the generalization performance and possible data imbalance, we calculate the weighted f1-score on the test sets,”.
>
> **References**
>
> [1] N. Cesa-Bianchi et al. “Worst-case analysis of selective sampling for linear classification”. JMLR (2016)
>
> [2] David Sculley et al. “Practical learning from one-sided feedback.” SIGKDD (2007)
>
> [3] https://scikit-learn.org/stable/modules/generated/sklearn.metrics.f1_score.html

---

### Official Review · Reviewer_osVX · 2024-03-24

**Q2-1 Originality-Novelty:** 2
**Q2-2 Correctness-Technical Quality:** 3
**Q2-5 Clarity Of Writing:** 3

**Q1 Summary And Contributions:**

This paper studies the online-based active learning (OAL) problem. Authors uses partial monitoring (PM) for OAL tasks and proposes NeuralCBP with deep neural networks for a class of cost-sentitive OAL tasks. Experiments validate the good performance of proposed method.

**Q2-3 Extent To Which Claims Are Supported By Evidence:**

3: Good: the main claims are supported by convincing evidence (in the form of adequate experimental evaluation, proofs, (pseudo-)code, references, assumptions).

**Q2-4 Reproducibility:**

3: Good: key resources (e.g. proofs, code, data) are available and key details (e.g. proofs, experimental setup) are sufficiently well-described for competent researchers to confidently reproduce the main results.

**Q3 Main Strengths:**

- Expanding the PM framework to multi-class and cost-sensitive OAL classification tasks.
- Proposed method expands existing method and works good in the experiments.
- Presentation is generally clear.

**Q4 Main Weakness:**

- The contributions over with existing theories and methods are somewhat limited.
- No theoretical analysis for NeuralCBP

**Q5 Detailed Comments To The Authors:**

The paper writing is generally clear. Since I am new to the PM framework, I would like the authors to clarify their contributions compared to the literature:

- the claimed contribution of generalizing binary label-efficient game to multi-class case seems not hard and somewhat straightforward.
- the NeuralCBP approach seems to directly use neural networks for estimates, to go beyond the linear and logistic assumptions in Bartók et al. [2012b]. Other than this, what are other contributions compared to the CBP method?
- as a suggestion, please provide full name for "CPB" and also briefly introduce it. This helps readers to better evaluate the novel part of NeuralCPB.

**Q9 Complying With Reviewing Instructions:**

Yes

---

> ### Author Rebuttal · Authors · 2024-04-05
>
> Thank you for your constructive comments, we are glad to address your questions and improve the paper.
>
> **Clarification on the contribution and the originality/novelty.**
>
> - **Connecting ideas in separate fields**: We hypothesize and validate that we can establish a novel, non-trivial, connection between the field of active learning and the PM framework.
> - **Methodological**: We show how a generalization of the label efficient game reduces to existing OAL tasks and enables the formulation of novel cost-sensitive OAL tasks.
> - **Algorithmic**: We propose NeuralCBP, the first neural partial monitoring (PM) strategy. Existing PM strategies are limited to the linear and logistic settings [12,14], which constitute a bottleneck towards the adoption of PM in practice. NeuralCBP presents algorithmic dynamics that differ from existing OAL strategies, which can be of independent interest to the OAL community.
> - **Empirical**: Our main results is that NeuralCBP competes with, and even outperforms, the current ICLR 2024 SOTA [17] in multiple binary and multi-class settings.
>
> Since the reviewer mentions little familiarity with the PM field, we would like to discuss the structure of this field to highlight the originality of our contribution. Despite its potential, PM has had little to no adoption within and beyond ML. Documented applied studies of PM are restricted to synthetic experiments [7,8,9]. We are the first to identify a meaningful PM application, highlighting the potential of the framework in practice, after nearly two decades of theoretical and algorithmic developments. To support further adoption, we make our study fully reproducible with open-source code for strategies and evaluation environments. Appendix A shows how to instantiate the complex formal definitions of the PM framework.
>
> **Technical difference between NeuralCBP and CBP**
>
> Thank you for asking for this clarification. The technical difference between NeuralCBP and CBP is discussed in Section 4. We have edited the manuscript to clarify the discussion:
>
> ”Property 2.3 shows that for any uninformative action $i \notin \mathcal I$, the learned feedback distribution is always $\pi_i(x_t) = 1$. In some PM games, most of the actions are uninformative, as is the case for generalized label-efficient games (presented in Section 3) where only expert actions are informative. Therefore, attributing learnable parameters to uninformative actions, as is in current CBP strategies [12, 14], turns out to be inefficient. In contrast, NeuralCBP attributes learnable parameters only to the informative actions in the game (see Definition 4.4). Restricting learnable parameters to the subset of informative actions is essential since NeuralCBP relies on the Explore-Exploit networks (EE-Nets) that require a shared representation for the actions. Including non-informative actions would cause overfitting, unstable learning, and increased complexity.”
>
> **Theoretical analysis of NeuralCBP.**
>
> Obtaining guarantees for NeuralCBP is attainable, but it is unrealistic to get the analysis ready in the time of the rebuttal. Regret upper bounds for CBP strategies in the linear and logistic setting exist [9,11]. Furthermore, NeuralCBP employs the Exploration Exploitation Networks (EE-nets), which are studied with the neural kernel tangent theorem [13].
>
> Obtaining guarantees for PM strategies in OAL is an interesting avenue for future work. Traditionally, OAL seeks distinct upper bounds on the generalization error and on the label complexity. However, PM strategies are natively analyzed with regret upper bounds. Although a regret upper bound would be interesting to the PM community, it would hardly compare with existing OAL guarantees.
>
> The scope of our work is to establish a non-trivial connection between the PM framework and OAL. Our empirical results show that this connection is promising, as demonstrated by the competitive empirical performance of NeuralCBP. The nature of our contribution calls for convincing empirical evidence to make an interesting proof of concept and raise attention to this particular connection. Some important advances in online learning arose from empirical works, as it was the case with [14] who highlighted the value of Thompson Sampling in bandits, and [1] who proposed the first neural bandits strategy.
>
>
> **Generalization of the label efficient game.**
>
> The generalization to the multi-class setting with uniform cost-structure is natural. Yet, no one has performed or discussed the extension in over two decades. The non-trivial part lies in the extension to the multi-class setting with a non-uniform cost structure, which requires game theoretical tools (cell decomposition, dominated actions) to define the structure of the cost matrix. Contemporary OAL literature omits references to the label efficient game [2, 6, 13], suggesting the dialogue between both fields is not well established which reinforces the value of our work.

---

### Meta-Review · Area_Chair_S21G · 2024-04-16

The submitted paper was reviewed by 3 knowledgeable reviewers all of which recommended acceptance of the paper. Based on my own reading of the paper I was more critical initially but a brief discussion with the reviewers about the shortcomings and strengths alleviated parts of my concerns. On the positive side, the online-based active learning (OAL) in the partial monitoring setting, is relevant to the online learning literature, and the proposed algorithm NeuralCBP achieves good empirical performance. On the downside, the theoretical contribution of this paper is not too strong (beyond connection partial monitoring and OAL) and the experiments need several clarifications and an extended discussion. Some concrete points:
* It is unclear how the authors derived the tuned variants of the algorithms (I don't see a description in the main text nor Appendix C; what was their methodology? Which range did they consider for the tuning?). For instance in Figure 3a, the tuned variants of the baselines seem to be mainly worse than the "official" variants.
* Furthermore, a more detailed analysis of the experimental results would be interesting to better understand the merits of the proposed algorithm. For instance, the properties of the proposed algorithm relative to the baselines regarding the f1-score for different expert query volumes are very different for different problems. What are the properties that cause the differences?
* Furthermore, there are some questionable statements, e.g., when talking about Figure 3a, the authors say "More importantly, the final regret of NeuralCBP scales appropriately with the difficulty of the dataset (CIFAR10 is the most challenging, followed by Fashion, then MNIST), further validating its adaptiveness to the learning difficulty." - wouldn't we expect a similar figure regarding NeuralCBP if we were choosing random samples for labeling simply because of the different complexity of the datasets? (if they had included a random sampling baseline we could be sure).

If this paper is accepted, the authors should carefully revise their paper in line with the reviewers' and the meta-reviewer's comments when preparing the camera-ready version.